

# The Carbon Dioxide Removal Model Intercomparison Project (CDR-MIP): Rationale and experimental protocol for CMIP6design

David P. Keller[1,*], Andrew Lenton[2,3], Vivian Scott[4], Naomi E. Vaughan[5], Nico Bauer[6], Duoying Ji[7], Chris D. Jones[8], Ben Kravitz[9], Helene Muri[10], Kirsten Zickfeld[11]

*Corresponding author email: dkeller@geomar.de
[1]GEOMAR Helmholtz Centre for Ocean Research Kiel, Germany
[2]CSIRO Oceans and Atmospheres, Hobart, Australia
[3]Antarctic Climate and Ecosystems Cooperative Research Centre, Hobart, Australia
[4]School of GeoSciences, University of Edinburgh
[5]Tyndall Centre for Climate Change Research, School of Environmental Sciences, University of East Anglia, Norwich, UK.
[6]Potsdam Institute for Climate Impact Research, Research Domain Sustainable Solutions, 14473 Potsdam, Germany
[7]College of Global Change and Earth System Science, Beijing Normal University, Beijing, China
[8] Met Office Hadley Centre, Exeter, UK,
[9]Atmospheric Sciences and Global Change Division, Pacific Northwest National Laboratory, Richland, WA, USA.
[10]Department of Geosciences, University of Oslo, Oslo, Norway.
[11]Department of Geography, Simon Fraser University, Burnaby, Canada

**Abstract**
The recent IPCC reports state that continued anthropogenic greenhouse gas
emissions are changing the climate, threatening "severe, pervasive and
irreversible" impacts. Slow progress in emissions reduction to mitigate climate
change is resulting in increased attention on what is called *Geoengineering*,
*Climate Engineering*, or *Climate Intervention* – deliberate interventions to counter
climate change that seek to either modify the Earth's radiation budget or remove
greenhouse gases such as $CO_2$ from the atmosphere. When focused on $CO_2$, the
latter of these categories is called Carbon Dioxide Removal (CDR). Future
emission scenarios that stay well below 2°C, and all emission scenarios that do
not exceed 1.5°C warming by the year 2100, require some form of CDR. At
present, there is little consensus on the climate impacts and atmospheric $CO_2$
reduction efficacy of the different types of proposed CDR. To address this need
the Carbon Dioxide Removal Model Intercomparison Project (or CDR-MIP) was
initiated. This project brings together models of the Earth system in a common
framework to explore the potential, impacts, and challenges of CDR. Here, we
describe the first set of CDR-MIP experiments, which are formally part of the 6th
Coupled Model Intercomparison Project (CMIP6). These experiments that are
designed to address questions concerning CDR-induced climate "reversibility",
the response of the Earth system to direct atmospheric $CO_2$ removal (direct air
capture and storage), and the CDR potential and impacts of
afforestation/reforestation, as well as ocean alkalinization.


## 1. Introduction

The Earth system is sensitive to the concentration of atmospheric greenhouse gases (GHG) because they have a direct impact on the planetary energy balance (Hansen, 2005), and in many cases also on biogeochemical cycling (IPCC, 2013). The concentration of one particularly important GHG, carbon dioxide ($CO_2$), has increased from approximately 277 ppm in the year 1750 to over 400 ppm today as a result of anthropogenic activities (Dlugokencky and Tans, 2016; Le Quéré et al., 2015). This $CO_2$ increase, along with other GHG increases and anthropogenic activities (e.g. land use change), has perturbed the Earth's energy balance leading to an observed global mean surface air temperature increase of around 0.8 °C above preindustrial (year 1850) levels in the year 2015 [updated from Morice et al. (2012)]. Biogeochemistry on land and in the ocean has also been affected by the increase in $CO_2$, with a well-observed decrease in ocean pH being one of the most notable results (Gruber, 2011; Hofmann and Schellnhuber, 2010). Many of the changes attributed to this rapid temperature increase and perturbation of the carbon cycle have been detrimental for natural and human systems (IPCC, 2014a).

While recent trends suggest that the atmospheric $CO_2$ concentration is likely to continue to increase (Peters et al., 2013; Riahi et al., 2017), the Paris Agreement of the 21st session of the Conference of Parties (COP21) on climate change (UNFCCC, 2016) has set the goal of limiting anthropogenic warming to well below 2°C (ideally no more than 1.5°C) relative to the global mean preindustrial temperature. To do this a massive climate change mitigation effort to reduce the sources or enhance the sinks of greenhouse gases (IPCC, 2014b) must be undertaken. Even if significant efforts are made to reduce $CO_2$ emissions, it will likely take decades before net emissions approach zero (Bauer et al., 2017; Riahi et al., 2017; Rogelj et al., 2015a), a level that is likely required to reach and maintain such temperature targets (Rogelj et al., 2015b). Changes in the climate will therefore continue for some time, with future warming strongly dependent on cumulative $CO_2$ emissions (Allen et al., 2009; IPCC, 2013; Matthews et al., 2009), and there is the possibility that "severe, pervasive and irreversible" impacts will occur if too much $CO_2$ is emitted (IPCC, 2013, 2014a).

The lack of agreement on how to sufficiently reduce $CO_2$ emissions in a timely
manner, and the magnitude of the task required to transition to a low carbon
world has led to increased attention on what is called *Geoengineering*, *Climate*
*Engineering*, or *Climate Intervention*.  These terms are all used to define actions
that deliberately manipulate of the climate system in an attempt to ameliorate or
reduce the impact of climate change by either modifying the Earth's radiation
budget (Solar Radiation Management, or SRM), or by removing the primary
greenhouse gas, $CO_2$, from the atmosphere (Carbon Dioxide Removal, or CDR)
(National Research Council, 2015).  In particular, there is an increasing focus and
study on the potential of carbon dioxide removal (CDR) methods to offset
emissions and eventually to enable "net negative emissions", whereby more $CO_2$
is removed via CDR than is emitted by anthropogenic activities, to complement
emissions reduction efforts.  CDR has also been proposed as a means of
"reversing" climate change if too much $CO_2$ is emitted, i.e., CDR may be able to
reduce atmospheric $CO_2$ to return radiative forcing to some target level.
All Integrated Assessment Model (IAM) scenarios of the future state that
some form of CDR will be needed to prevent the mean global surface
temperature from exceeding 2°C (Bauer et al., 2017; Fuss et al., 2014; Kriegler et
al., 2016; Rogelj et al., 2015a).  Most of these limited warming scenarios feature
overshoots in radiative forcing around mid-century, which is closely related to
the amount of cumulative CDR up until the year 2100 (Kriegler et al., 2013).
Despite the prevalence of CDR in these scenarios, and its increasing utilization in
political and economic discussions, many of the methods by which this would be
achieved at this point rely on immature technologies (National Research Council,
2015; Schäfer et al., 2015). Large scale CDR methods are not yet a commercial
product, and hence questions remain about their feasibility, realizable potential
and risks (Smith et al., 2015; Vaughan and Gough, 2016).
Overall, knowledge about the potential climatic, biogeochemical,
biogeophysical, and other impacts in response to CDR is still quite limited, and
large uncertainties remain, making it difficult to comprehensively evaluate the
potential and risks of any particular CDR method and make comparisons
between methods.  This information is urgently needed to allow us to assess:

i.    The degree to which CDR could help mitigate or perhaps reverse climate
change;


ii.    The potential risks/benefits of different CDR proposals; and

iii.    To inform how climate and carbon cycle responses to CDR could be
included when calculating and accounting for the contribution of CDR in
mitigation scenarios, i.e., so that CDR is better constrained when it is
included in IAM generated scenarios.


To date, modelling studies of CDR focusing on the carbon cycle and
climatic responses have been undertaken with only a few Earth system models
(Arora and Boer, 2014; Boucher et al., 2012; Cao and Caldeira, 2010; Gasser et al.,
2015; Jones et al., 2016a; Keller et al., 2014; MacDougall, 2013; Mathesius et al.,
2015; Tokarska and Zickfeld, 2015; Zickfeld et al., 2016).  However, as these
studies all use different experimental designs, their results are not directly
comparable, consequently building a consensus on responses is challenging.  A
model intercomparison study with Earth System Models of Intermediate
Complexity (EMICS) that addresses climate reversibility, among other things, has
recently been published (Zickfeld et al., 2013), but the focus was on the very
distant future rather than this century.  Moreover, in many of these studies,
atmospheric $CO_2$ concentrations were prescribed rather than being driven by
$CO_2$ emissions and thus, the projected changes were independent of the strength
of feedbacks associated with the carbon cycle.
Given that Earth system models are one of the few tools available for
making quantifications at these scales, as well as for making projections into the
future, CDR assessments must include emissions-driven modeling studies to
capture the carbon-cycle feedbacks.  However, such an assessment cannot be
done with one or two models alone, since this will not address uncertainties due
to model structure and internal variability. Below we describe the scientific foci
and several experiments (Table 1) that comprise the initial phase of the CMIP6
endorsed Carbon Dioxide Removal Model Intercomparison Project (CDR-MIP).


## 1.2 CDR-MIP Scientific Foci



There are three principal science motivations behind CDR-MIP. First and
foremost, CDR-MIP will provide information that can be used to help assess the
potential and risks of using CDR to address climate change. A thorough
assessment will need to look at both the impacts of CDR upon the Earth system
and human society. CDR-MIP will focus primarily on Earth system impacts, with
the anticipation that this information will also be useful for understanding
potential impacts upon society. The scientific outcomes will lead to more
informed decisions about the role CDR may play in climate change mitigation
(defined here as a human intervention to reduce the sources or enhance the
sinks of greenhouse gases). CDR-MIP experiments will also provide an
opportunity to better understand how the Earth system responds to
perturbations, which is relevant to many of the Grand Science Challenges posed
by the World Climate Research Program (WCRP; https://www.wcrp-
climate.org/grand-challenges/grand-challenges-overview). CDR-MIP
experiments provide a unique opportunity because the perturbations are often
opposite in sign to previous CMIP perturbation experiments ($CO_2$ is removed
instead of added). Second, CDR-MIP results may also be able to provide
information that helps to understand how model resolution and complexity
cause systematic model bias. In this instance, CDR-MIP experiments may be
especially useful for gaining a better understanding of the similarities and
differences between global carbon cycle models because we invite a diverse
group of models to participate in CDR-MIP. Finally, CDR-MIP results can help to
quantify uncertainties in future climate change scenarios, especially those that
include CDR. In this case CDR-MIP results may be useful for calibrating CDR
inclusion in IAMs during the scenario development process.
The initial foci that are addressed by CDR-MIP include (but are not limited
to):

(i) Climate "reversibility": assessing the efficacy of using CDR to return high
future atmospheric $CO_2$ concentrations to lower levels. This topic is highly
idealized, as the technical ability of CDR methods to remove such enormous
quantities of $CO_2$ on relatively short timescales (i.e., this century) is doubtful.
However, the results will provide information on the degree to which a changing
and changed climate could be returned to a previous state.  This knowledge is
especially important since socio-economic scenarios that limit global warming to
well below 2° C often feature radiative forcing overshoots that must be
"reversed" using CDR. Specific questions on reversibility will address:

1)     What components of the Earth's climate system exhibit "reversibility"

when $CO_2$ increases and then decreases?  On what timescales do these

"reversals" occur?  And if reversible, is this complete reversibility or

just on average (are there spatial and temporal aspects)?

2)     Which, if any, changes are irreversible?
3)     What role does hysteresis play in these responses?

(ii) The potential efficacy, feedbacks, and side effects of specific CDR methods.
Efficacy is defined here as $CO_2$ removed from the atmosphere, over a specific
time horizon, as a result of a specific unit of CDR action. This topic will help to
better constrain the carbon sequestration potential and risks and/or benefits of
selected methods.  Together, a rigorous analysis of the nature, sign, and
timescales of these CDR-related topics will provide important information for the
inclusion of CDR in climate mitigation scenarios, and in resulting mitigation and
adaptation policy strategies. Specific questions on individual CDR methods will
address:

1)     How much $CO_2$ would have to be removed to return to a specified

concentration level e.g. present day or pre-industrial?

2)     What are the short-term carbon cycle feedbacks (e.g. rebound)

associated with the method?

3)     What are the short- and longer-term physical/chemical/biological

impacts and feedbacks, and potential side effects of the method?

4)     For methods that enhance natural carbon uptake, e.g., afforestation

or ocean alkalinization, where is the carbon stored (land and

ocean) and for how long (i.e. issues of permanence; at least as

much as this can be calculated with these models)?


**1.3 Structure of this document**

Our motivation for preparing this document is to lay out in detail the

CDR-MIP experimental protocol, which we request all modelling groups to follow
as closely as possible. Firstly, in Section 2, we review the scientific background
and motivation for CDR in more detail than covered in this introduction. Section
3 describes some requirements and recommendations for participating in CDR-
MIP and describes links to other CMIP6 activities. Section 4 describes each CDR-
MIP simulation in detail. Section 5 describes the model output and data policy.
Section 6 presents an outlook of potential future CDR-MIP activities and a
conclusion.  Section 7 describes how to obtain the model code and data used
during the production of this document.

**2. Background and motivation**

At present, there are two main proposed CDR approaches, which we

briefly introduce here. The first category encompasses methods that are
primarily designed to enhance the Earth's natural carbon sequestration
mechanisms. Enhancing natural oceanic and terrestrial carbon sinks is suggested
because these sinks have already *each* taken up over a quarter of the carbon
emitted as a result of anthropogenic activities (Le Quéré et al., 2016) and have
the capacity to store additional carbon, although this is subject to environmental
limitations.  Some prominent proposed sink enhancement methods include
afforestation or reforestation, enhanced terrestrial weathering, biochar, land
management to enhance soil carbon storage, ocean fertilization, ocean
alkalinization, and coastal management of blue carbon sinks.

The second general CDR category includes methods that rely primarily on

technological means to directly remove carbon from the atmosphere, ocean, or
land and isolate it from the climate system, e.g., storage in a geological reservoir
(Scott et al., 2015).  Methods that are primarily technological are suggested
because they may not be as limited by environmental constraints.  Some
prominent proposed technological methods include direct $CO_2$ air capture with
storage and seawater carbon capture (and storage).  One other proposed CDR
method, bioenergy with carbon capture and storage (BECCS), relies on both
natural processes and technology.  BECCS is thus, constrained by some
environmental limitations (e.g., suitable land area), but because the carbon is
removed and ultimately stored elsewhere, it may have a higher CDR potential
than if the same deployment area were used for a sink-enhancing CDR method
like afforestation that stores carbon permanently above ground and reaches a
saturation level for a given area (Smith et al., 2015).

From an Earth system perspective, the potential and impacts of proposed

CDR methods have only been investigated in a few individual studies - see recent
climate intervention assessments for a broad overview of the state of CDR
research (National Research Council, 2015; Rickels et al., 2011; The Royal
Society, 2009; Vaughan and Lenton, 2011) and references therein. These studies
agree that CDR application at a large scale ($\geq 1$Gt $CO_2$ yr$^{-1}$) would likely have a
substantial impact on the climate, biogeochemistry and the ecosystem services
that the Earth provides (i.e., the benefits humans obtain from ecosystems)
(Millennium Ecosystem Assesment, 2005).  Idealized Earth system model
simulations suggest that CDR does appear to be able to limit or even reverse
warming and changes in many other key climate variables (Boucher et al., 2012;
Tokarska and Zickfeld, 2015; Wu et al., 2014; Zickfeld et al., 2016).  However,
less idealized studies, e.g., when some environmental limitations are accounted
for, suggest that many methods have only a limited individual mitigation
potential (Boysen et al., 2016, 2017; Keller et al., 2014; Sonntag et al., 2016).

Studies have also focused on the carbon cycle response to the deliberate

redistribution of carbon between dynamic carbon reservoirs or permanent
(geological) carbon removal. Understanding and accounting for the feedbacks
between these reservoirs in response to CDR is particularly important for
understanding the efficacy of any method (Keller et al., 2014). For example,
when $CO_2$ is removed from the atmosphere in simulations, the rate of oceanic
$CO_2$ uptake, which has historically increased in response to increasing emissions,
is reduced and might eventually reverse (i.e., net outgassing), because of a
reduction in the air-sea flux disequilibrium (Cao and Caldeira, 2010; Jones et al.,
2016a; Tokarska and Zickfeld, 2015; Vichi et al., 2013). Equally, the terrestrial
carbon sink also weakens in response to atmospheric $CO_2$ removal, and can also
become a source of $CO_2$ to the atmosphere (Cao and Caldeira, 2010; Jones et al.,
2016a; Tokarska and Zickfeld, 2015). This 'rebound' carbon flux response that
weakens or reverses carbon uptake by natural carbon sinks would oppose CDR
and needs to be accounted for if the goal is to limit or reduce atmospheric $CO_2$
concentrations to some specified level (IPCC, 2013).
In addition to the climatic and carbon cycle effects of CDR, most methods
appear to have side effects (Keller et al., 2014). The impacts of these side effects
tend to be method specific and may amplify or reduce the climate change
mitigation potential of the method. Some significant side effects are caused by
the spatial scale (e.g., millions of $km^2$) at which many methods would have to be
deployed to have a significant impact upon $CO_2$ and global temperatures (Boysen
et al., 2016; Heck et al., 2016; Keller et al., 2014). Side effects can also potentially
alter the natural environment by disrupting biogeochemical and hydrological
cycles, ecosystems, and biodiversity (Keller et al., 2014). For example, large-
scale afforestation could change regional albedo and evapotranspiration and so
have a biogeophysical impact on the Earth's energy budget and climate (Betts,
2000; Keller et al., 2014). Additionally, if afforestation were done with non-
native plants or monocultures to increase carbon removal rates this could impact
local biodiversity. For human societies, this means that CDR-related side effects
could potentially impact the ecosystem services provided by the land and ocean
(e.g., food production), with the information so far suggesting that there could be
both positive and negative impacts on these services. Such effects could change
societal responses and strategies for climate change adaptation if large-scale
CDR were to be deployed.
CDR deployment scenarios have focused on both preventing climate
change and reversing it. While there is some understanding of how the Earth
system may respond to CDR, as described above, another dynamic comes into
play if CDR were to be applied to "reverse" climate change. This is because if
CDR were deployed for this purpose, it would deliberately change the climate,
i.e., drive it in another direction, rather than just prevent it from changing by

limiting $CO_2$ emissions. Few studies have investigated how the Earth system may respond if CDR is applied in this manner. The link between cumulative $CO_2$ emissions and global mean surface air temperature change has been extensively studied (IPCC, 2013). Can this change simply be reversed by removing the $CO_2$ that has been emitted since the preindustrial era? Little is known about how reversible this relationship is, or whether it applies to other Earth system properties (e.g., net primary productivity, sea level, etc.).  Investigations of CDR-induced climate reversibility have suggested that many Earth system properties are "reversible", but often with non-linear responses (Armour et al., 2011; Boucher et al., 2012; MacDougall, 2013; Tokarska and Zickfeld, 2015; Wang et al., 2014; Wu et al., 2014; Zickfeld et al., 2016).  However, these analyses were generally limited to global annual mean values, and most models did not include potentially important components such as permafrost or terrestrial ice sheets. Thus, there are many unknowns and much uncertainty about whether it is possible to "reverse" climate change.  Obtaining knowledge about climate "reversibility" is especially important as it could be used to direct or change societal responses and strategies for adaptation and mitigation.

**2.1 Why a model intercomparison study on CDR?**

Although ideas for controlling atmospheric $CO_2$ concentrations were proposed in the middle of the last century, it is only recently that CDR methods have received widespread attention as climate intervention strategies (National Research Council, 2015; Schäfer et al., 2015; The Royal Society, 2009; Vaughan and Lenton, 2011). While some proposed CDR methods do build upon substantial knowledge bases (e.g., soil and forest carbon, and ocean biogeochemistry), little research into large scale CDR has been conducted and limited research resources applied (National Research Council, 2015; Oschlies and Klepper, 2017).  The small number of existing laboratory studies and small-scale field trials of CDR methods were not designed to evaluate climate or carbon cycle responses to CDR. At the same time it is difficult to conceive how such an investigation could be carried out without scaling a method up to the point where it would essentially be "deployment". The few natural analogues that exist

for some methods (e.g., weathering or reforestation) only provide limited insight
into the effectiveness of deliberate large scale CDR. As such, beyond syntheses of
resource requirements and availabilities, e.g., Smith, (2016), there is a lack of
observational constraints that can be applied to the assessment of the
effectiveness of CDR methods. Lastly, many proposed CDR methods are pre-
mature at this point and technology deployment strategies would be required to
overcome this barrier (Schäfer et al., 2015), which means that they can only be
studied in an idealized manner, i.e., through model simulations.

Understanding the response of the Earth system to CDR is urgently
needed because CDR is increasingly being utilized to inform policy and economic
discussions. Examples of this include scenarios that are being developed with
GHG emission forcing that exceeds (or overshoots) what is required to limit
global mean temperatures to 2° C or 1.5 °C, with the assumption that
reversibility is possible with the future deployment of CDR. These scenarios are
generated using Integrated Assessment Models, which compute the emissions of
GHGs, short-lived climate forcers, and land-cover change associated with
economic, technological and policy drivers to achieve climate targets. Most
integrated assessment models represent BECCS as the only CDR option, with
only a few also including afforestation (IPCC, 2014b). During scenario
development and calibration the output from the IAMs is fed into climate models
of reduced complexity, e.g., MAGICC (Model for the Assessment of Greenhouse-
gas Induced Climate Change) (Meinshausen et al., 2011), to calculate the global
mean temperature achieved through the scenario choices, e.g., those in the
Shared Socioeconomic Pathways (SSPs) (Riahi et al., 2017). These climate
models are calibrated to Earth system models or based on modelling
intercomparison exercises like the Coupled Model Intercomparison Phase 5
(CMIP5), where much of the climate-carbon cycle information comes from the
Coupled Climate-Carbon Cycle Model Intercomparison Project (C4MIP).
However, since the carbon cycle feedbacks of large-scale negative $CO_2$ emissions
have not been explicitly analyzed in projects like CMIP5, with the exception of
Jones et al. (2016a), many assumptions have been made about the effects of CDR
on the carbon cycle and climate.  Knowledge of these short-term carbon cycle
feedbacks is needed to better constrain the effectiveness of the CDR technologies
assumed in the IAM generated scenarios.
This relates to the policy relevant question of whether in a regulatory
framework, $CO_2$ removals from the atmosphere should be treated like emissions
except for the opposite (negative) sign or if specific methods, which may or may
not have long-term consequences (e.g., afforestation/reforestation vs. direct $CO_2$
air capture with geological carbon storage), should be treated differently. The
lack of this kind of analyses is a knowledge gap in current climate modeling
(Jones et al., 2016a) and relevant for IAM models and political decisions. There is
an urgent need to close this gap since additional CDR options like the enhanced
weathering of rocks on land or direct air capture continue to be included in IAMs,
e.g., Chen and Tavoni (2013). For the policy relevant questions it is also
important to analyze the carbon cycle effects given realistic policy scenarios
rather than idealized perturbations.

**3. Requirements and recommendations for participation in CDR-MIP**

The CDR-MIP initiative is designed to bring together a suite of Earth
System Models, Earth System Models of Intermediate Complexity (EMICs), and
potentially even box models in a common framework. Note that only models
that meet certain requirements (https://pcmdi.llnl.gov/CMIP6/Guide/) can
participate in an official CMIP6 capacity. Models of differing complexities are
invited to participate because the questions posed above cannot be answered
with any single class of models. For example, ESMs are primarily suited for
investigations spanning only the next century because of the computational
expense, while EMICs and box models are well suited to investigate the long-
term questions surrounding CDR, but are often highly parameterized and may
not include important processes, e.g., cloud feedbacks. The use of differing
models will also provide insight into how model resolution and complexity
controls modeled short- and long-term climate and carbon cycle responses to
CDR.
All groups that are running models with an interactive carbon cycle are
encouraged to participate in CDR-MIP. We desire diversity and encourage groups
to use older models, with well-known characteristics, biases and established
responses (e.g. previous CMIP model versions), as well as state-of-the-art CMIP6
models. For longer model simulations, we would encourage modellers when
possible to include additional carbon reservoirs, such as ocean sediments or
permafrost, as these are not always implemented for short simulations. Models
that only include atmospheric and oceanic carbon reservoirs are welcome, and
will be able to participate in some experiments. All models wishing to participate
in CDR-MIP must provide clear documentation that details the model version,
components, and key run-time and initialization information (model time
stepping, spin-up state at initialization, etc.). Furthermore, all model output must
be standardized to facilitate analyses and public distribution (see Sections 4 and

446 5).


**3.1 Relations to other MIPs**

There are no existing MIPs with experiments focused on climate
"reversibility", direct $CO_2$ air capture (with storage), or ocean alkalinization.
However, this does not mean that there are no links between CDR-MIP and other
MIPs. CMIP6 and CMIP5 experiments, analyses, and assessments both provide a
valuable baseline and model sensitivities that can be used to better understand
CDR-MIP results and we highly recommend that participants in CDR-MIP also
conduct other MIP experiments. Further, to maximize the use of computing
resources CDR-MIP uses experiments from other MIPs as a control run for a
CDR-MIP experiment or to provide a pathway from which a CDR-MIP experiment
branches (Sections 3.2 and 4, Tables 2- 7). Principle among these is the CMIP
Diagnostic, Evaluation, and Characterization of Klima (DECK) and historical
experiments as detailed in Eyring et al. (2016) for CMIP6, since they provide the
basis for many experiments with almost all MIPs leveraging these in some way.
Here, we additionally describe links to ongoing MIPs that are endorsed by
CMIP6, noting that earlier versions of many of these MIPs were part of CMIP5
and so provide a similar synergy for any CMIP5 models participating in CDR-MIP.
Given the emphasis on carbon cycle perturbations in CDR-MIP, there is a
strong synergy with C4MIP which provides a baseline, standard protocols, and
diagnostics for better understanding the relationship between the carbon cycle
and the climate in CMIP6 (Jones et al., 2016b). For example, the C4MIP
emissions-driven SSP5-8.5 scenario (a high $CO_2$ emission scenario with a
radiative forcing of 8.5 Wm$^{-2}$ in year 2100) simulation, *esm-ssp585*, is a control
run and branching pathway for several CDR-MIP experiments. CDR-MIP
experiments may equally be valuable for understanding model responses during
related C4MIP experiments. For example, the C4MIP experiment *ssp534-over-bgc*
is a concentration driven "overshoot" scenario simulation that is run in a
partially coupled mode. The simulation required to analyze this experiment is a
fully coupled $CO_2$ concentration driven simulation of this scenario, *ssp534-over,*
from the Scenario Model Intercomparison Project (ScenarioMIP). The novel CDR-
MIP experiment, *CDRC2--overshoot*, which is a fully coupled $CO_2$ emission driven
version of this scenario, will provide additional information that can be used to
extend the analyses to better understand climate-carbon cycle feedbacks.

The Land Use Model Intercomparison Project (LUMIP) is designed to

better understand the impacts of land-use and land-cover change on the climate
(Lawrence et al., 2016). The three main LUMIP foci overlap with some of the
CDR-MIP foci, especially in regards to land management as a CDR method (e.g.,
afforestation/reforestation). To facilitate land-use and land-cover change
investigations LUMIP provides standard protocols and diagnostics for the
terrestrial components of CMIP6 Earth system models. The inclusion of these
diagnostics will be important for all CDR-MIP experiments performed with
CMIP6 models. The CDR-MIP experiment on afforestation/reforestation, *CDRC3-*
*afforestation* (*esm-ssp585-ssp126Lu-ext)*, is also an extension of the LUMIP *esm-*
*ssp585-ssp126Lu* simulation beyond 2100 to investigate the long-term
consequences of afforestation/reforestation in a high-$CO_2$ world (Section 4.3).

ScenarioMIP is designed to provide multi-model climate projections for

several scenarios of future anthropogenic emissions and land use changes
(O'Neill et al., 2016), and provides baselines or branching for many MIP
experiments . The ScenarioMIP SSP5-3.4-OS experiments, *ssp534-over* and
*ssp534-over-ext*, which prescribe atmospheric $CO_2$ to follow an emission
overshoot pathway that is followed by aggressive mitigation to reduce emissions
to zero by about 2070, with substantial negative global emissions thereafter, are

**Formatted:** Font: Italic

**Formatted:** Font: Italic

used as control runs for the CDR-MIP $CO_2$ emission driven version of this
scenario.  Along with the partially coupled C4MIP version of this experiment,
these experiments will allow for qualitative comparative analyses to better
understand climate-carbon cycle feedbacks in an "overshoot" scenario with
negative emissions (CDR). If it is found that the carbon cycle effects of CDR are
improperly accounted for in the scenarios, then this information can be used to
recalibrate older CDR-including IAM scenarios and be used to better constrain
CDR when it is included in new scenarios.
The Ocean Model Intercomparison Project (OMIP), which primarily
investigates the ocean-related origins and consequences of systematic model
biases, will help to provide an understanding of ocean component functioning for
models participating in CMIP6 (Griffies et al., 2016). OMIP will also establish
standard protocols and output diagnostics for ocean model components. The
biogeochemical protocols and diagnostics of OMIP (Orr et al., 2016) are
particularly relevant for CMIP6 models participating in CDR-MIP. While the
inclusion of these diagnostics will be important for all CDR-MIP experiments,
these standards will be particularly important for facilitating the analysis of our
marine CDR experiment on ocean alkalinization, *CDRC4-ocean-alk* (Section 4.4).
**3.2 Prerequisite and recommended CMIP simulations**
The following CMIP experiments are considered prerequisites for
specified CDR-MIP experiments (Tables 2- 7) and analyses:
- The CMIP prescribed atmospheric $CO_2$ pre-industrial control simulation,
*piControl.* This is required for all CDR-MIP experiments (many control
runs and experiment prerequisites branch from this) and is usually done
as part of the spin-up process.
- The CMIP6 pre-industrial control simulation with interactively simulated
atmospheric $CO_2$ (i.e., the $CO_2$ concentration is internally calculated, but
emissions are zero), *esm-piControl.* This is required for CDR-MIP

experiments *CDR~~C2~~-pi-pulse, CDR~~C2~~-overshoot~~, C3~~, CDR-afforestation* 
and *CDR-ocean-alk~~C4~~*.

- The CMIP 1 % per year increasing $CO_2$ simulation, *1pctCO2*, that is initialized from a pre-industrial $CO_2$ concentration with $CO_2$ then increasing by 1% per year until the $CO_2$ concentration has quadrupled (approximately 139 years). This is required for CDR-MIP experiment *CDR~~C1~~-reversibility*.

- The CMIP6 historical simulation, *historical*, where historical atmospheric $CO_2$ forcing is prescribed along with land use, aerosols, and non-$CO_2$ greenhouse gases forcing. This is required for CDR-MIP experiment *CDR~~C2~~-yr2010-pulse*.

- The CMIP6 emissions driven historical simulation, *esm-hist*, where the atmospheric $CO_2$ concentration is internally calculated in response to historical anthropogenic $CO_2$ emissions forcing. Other forcing such as land use, aerosols, and non-$CO_2$ greenhouse gases are prescribed.  This is required for CDR-MIP experiments *C~~DR2~~-overshoot, C~~DR3~~-afforestation*, and *C~~DR4~~-ocean-alk*.

- The LUMIP *esm-ssp585-ssp126Lu* simulation, which simulates afforestation in a high $CO_2$ emission scenario, is the basis for CDR-MIP experiment *esm-ssp585-ssp126Lu-ext*.

- The C4MIP *esm-ssp585* simulation, which is a high emission scenario and serves as a control run and branching pathway for CDR-MIP *CDR4-ocean-alk* experiment.

We also highly recommend that groups run these additional C4MIP and ScenarioMIP simulations:

- The ScenarioMIP *ssp534-over* and *ssp534-over-ext* simulations, which prescribe the atmospheric $CO_2$ concentration to follow an emission overshoot pathway that is followed by aggressive mitigation to reduce emissions to zero by about 2070, with substantial negative global emissions thereafter. These results can be qualitatively compared to CDR-MIP experiment *CDR2--overshoot,* which is the same scenario, but driven by $CO_2$ emissions.

- The C4MIP *ssp534-over-bgc* and *ssp534-over-bgcExt* simulations, which are biogeochemically-coupled versions of the *ssp534-over* and *ssp534-over-ext* simulations, i.e., only the carbon cycle components (land and ocean) see the prescribed increase in the atmospheric $CO_2$ concentration; the model's radiation scheme sees a fixed preindustrial $CO_2$ concentration. These results can be qualitatively compared to CDR-MIP experiment *CDR2--overshoot,* which is a fully coupled version of this scenario.

**3.3 Simulation ensembles**

We encourage participants whose models have internal variability to conduct multiple realizations, i.e. ensembles, for all experiments. While these are highly desirable, they are neither mandatory, nor a prerequisite for participation in CDR-MIP. Therefore, the number of ensemble members is at the discretion of each modeling group.  However, we strongly encourage groups to submit at least three ensemble members if possible.

**3.4 Climate sensitivity calculation**

Knowing the climate sensitivity of each model participating in CDR-MIP is important for interpreting the results. For modelling groups that have not already calculated their model's climate sensitivity, the required CMIP *1pctCO2* simulation can be used to calculate both the transient and equilibrium climate

sensitivities. The transient climate sensitivity can be calculated as the difference
in the global annual mean surface temperature between the start of the
experiment and a 20-year period centered on the time of $CO_2$ doubling. The
equilibrium response can be diagnosed following Gregory et al. (2004), Frölicher
et al. (2013), or if possible (desirable) by running the model to an equilibrium
state at $2\times CO_2$ or $4\times CO_2$.

**3.5 Model drift**

Model drift (Gupta et al., 2013; Séférian et al., 2015) is a concern for all
CDR-MIP experiments because if a model is not at an equilibrium state when the
experiment or prerequisite CMIP experiment begins, then the response to any
experimental perturbations could be confused by drift. Thus, before beginning
any of the experiments a model must be spun-up to eliminate long-term drift in
carbon reservoirs or fluxes. Groups participating in CMIP6 should follow the
C4MIP protocols described in Jones et al. (2016b), to ensure that drift is
acceptably small. This means that land, ocean and atmosphere carbon stores
should each vary by less than 10 GtC per century (long-term average $\leq 0.1$ Gt C
$yr^{-1}$). We leave it to individual groups to determine the length of the run required
to reach such a state. If older model versions, e.g., CMIP5, are used for any
experiments, any known drift should be documented.

**4. Experimental Design and Protocols**

To facilitate multiple model needs, the experiments described below have
been designed to be relatively simple to implement. In most cases, they were also
designed to have high signal-to-noise ratios to better understand how the
simulated Earth system responds to significant CDR perturbations. While there
are many ways in which such experiments could be designed to address the
questions surrounding climate reversibility and each proposed CDR method, the
CDR-MIP like all MIPs, must be limited to a small number of practical
experiments. Therefore, after careful consideration, one experiment was chosen
specifically to address climate reversibility and several more were chosen to
investigate CDR by idealized direct air capture of $CO_2$ (DAC),
afforestation/reforestation, and ocean alkalinization (Table 1). Experiments are
prioritized based on a tiered system, although, we encourage modelling groups
to complete the full suite of experiments. Unfortunately, limiting the number of
experiments means that a number of potentially promising or widely utilized
CDR methods or combinations of methods must wait until a later time, i.e., a 2nd
phase, to be investigated in a multi-model context. In particular, the exclusion of
Biomass Energy with Carbon Capture and Storage (BECCS) is unfortunate, as this
is the primary CDR method in the Representative Concentration Pathways (RCP)
and Shared Socio-economic Pathways (SSP) scenarios used in CMIP5 and 6,
respectively.  However, there was no practical way to design a less idealized
BECCS experiment as most state-of-the-art models are either incapable of
simulating a biomass harvest with permanent removal or would require a
substantial amount of reformulating to do so in a manner that allows comparable
multi-model analyses.
In some of the experiments described below we ask that non-$CO_2$ forcing
(e.g., land use change, radiative forcing from other greenhouse gases, etc.) be
held constant, e.g. at that of a specific year, so that only changes in other forcing,
like $CO_2$ emissions, drive the main model response.  For some forcing, e.g. aerosol
emissions, this may mean that monthly changes in forcing are repeated
throughout the rest of the simulation as if it was always one particular year.
However, we recognize that models apply forcing in different ways and leave it
to individual modelling groups to determine the best way hold forcing constant.
We request that the methodology for holding forcing constant be documented
for each model.
**4.1. Climate and carbon cycle reversibility experiment (_CDR1-reversibility_)**
If $CO_2$ emissions are not reduced quickly enough, and more warming
occurs than is desirable or tolerable, then it is important to understand if CDR
has the potential to "reverse" climate change. Here we propose an idealized Tier
1 experiment that is designed to investigate CDR-induced climate "reversibility"
(Fig. 1, Table 2). This experiment investigates the "reversibility" of the climate
system by leveraging the prescribed 1% yr$^{-1}$ CO$_2$ concentration increase
experiment that was done for prior CMIPs, and is a key run for CMIP6 (Eyring et
al., 2016; Meehl et al., 2014). The CDR-MIP experiment starts from the 1% yr$^{-1}$
CO$_2$ concentration increase experiment, *1pctCO2*, and then at the 4×CO$_2$
concentration level prescribes a -1% yr$^{-1}$ removal of CO$_2$ from the atmosphere to
pre-industrial levels [Fig. 1; this is also similar to experiments in Boucher et al.,
(2012) and Zickfeld et al., (2016)].   This approach is analogous to an unspecified
CDR application or DAC, where CO$_2$ is removed to permanent storage to return
atmospheric CO$_2$ to a prescribed level, i.e., a preindustrial concentration. To do
this, CDR would have to counter emissions (unless they have ceased) as well as
changes in atmospheric CO$_2$ due to the response of the ocean and terrestrial
biosphere.  We realize that the technical ability of CDR methods to remove such
enormous quantities of CO$_2$ on such a relatively short timescale (i.e. in a few
centuries) is unrealistic. However, branching from the existing CMIP *1pctCO2*
experiment provides a relatively straightforward opportunity, with a high signal-
to-noise ratio, to explore the effect of large-scale removal of CO$_2$ from the
atmosphere and issues involving reversibility (Fig. 2 shows exemplary *CDR1-*
*reversibility* results from two models).

**4.1.1 Protocol for *CDR1-reversibility***

*Prerequisite simulations:*  Perform the CMIP *piControl* and the *1pctCO2*
experiments. The *1pctCO2* experiment branches from the DECK *piControl*
experiment, which should ideally represent a near-equilibrium state of the
climate system under imposed year 1850 conditions. Starting from year 1850
conditions (*piControl* global mean atmospheric CO$_2$ should be 284.7 ppm) the
*1pctCO2* simulation prescribes a CO$_2$ concentration increase at a rate of 1% yr$^{-1}$
(i.e., exponentially). The only externally imposed difference from the *piControl*
experiment is the change in CO$_2$, i.e., all other forcing is kept at that of year 1850.
A restart must be generated when atmospheric CO$_2$ concentrations are four
times that of the *piControl* simulation (1138.8 ppm; this should be 140 years into
the run). Groups that have already performed the *piControl* and *1pctCO2*
simulations for CMIP5 or CMIP6 may provide a link to them if they are already
on the Earth System Grid Federation (ESGF) that host CMIP data.

*1pctCO2-cdr* simulation: Use the 4×$CO_2$ restart from *1pctCO2* and prescribe a 1%
$yr^{-1}$ removal of $CO_2$ from the atmosphere (start removal at the beginning of the
140$^{th}$ year: January 1$^{st}$.) until the $CO_2$ concentration reaches 284.7 ppm (140
years of removal).  As in *1pctCO2* the only externally imposed forcing should be
the change in $CO_2$ (all other forcing is kept at that of year 1850). The $CO_2$
concentration should then be held at 284.7 ppm for as long as possible (a
minimum of 60 years is required), with no change in other forcing. EMICs and
box models are encouraged to extend runs for at least 1000 years (and up to
5000 years) at 284.7 ppm $CO_2$ to investigate long-term climate system and
carbon cycle reversibility (see Fig. 2 b and d for examples of why it is important
to understand the long-term response).

**4.2 Direct $CO_2$ air capture with permanent storage experiments (*CDR~~2~~-pi-**
**pulse, CDR-year2010-pulse, CDR-overshoot*)**

The idea of directly removing excess $CO_2$ from the atmosphere (i.e.,

concentrations above pre-industrial levels) and permanently storing it in some
reservoir, such as a geological formation, is appealing because such an action
would theoretically address the main cause of climate change, anthropogenically
emitted $CO_2$ that remains in the atmosphere. Laboratory studies and small-scale
pilot plants have demonstrated that atmospheric $CO_2$ can be captured by several
different methods that are often collectively referred to as Direct Air Capture
(DAC) technology (Holmes and Keith, 2012; Lackner et al., 2012; Sanz-Pérez et
al., 2016). Technology has also been developed that can place captured carbon in
permanent reservoirs, i.e., Carbon Capture and Storage (CCS) methods (Matter et
al., 2016; Scott et al., 2013, 2015) . DAC technology is currently prohibitively
expensive to deploy at large scales and may be technically difficult to scale up
(National Research Council, 2015), but does appear to be a potentially viable
CDR option. However, aside from the technical questions involved in developing
and deploying such technology, there remain questions about how the Earth
system would respond if $CO_2$ were removed from the atmosphere.
Here we propose a set of experiments that are designed to investigate and
quantify the response of the Earth system to idealized large-scale DAC. In all
experiments, atmospheric $CO_2$ is allowed to freely evolve to investigate carbon
cycle and climate feedbacks in response to DAC. The first two idealized
experiments described below use an instantaneous (*pulse*) $CO_2$ removal from the
atmosphere - approach for this investigation. Instantaneous $CO_2$ removal
perturbations were chosen since *pulsed* $CO_2$ addition experiments have already
been proven useful for diagnosing carbon cycle and climate feedbacks in
response to $CO_2$ perturbations. For example, previous positive $CO_2$ pulse
experiments have been used to calculate Global Warming Potential (GWP) and
Global Temperature change Potential (GTP) metrics (Joos et al., 2013). The
experiments described below build upon the previous positive $CO_2$ pulse
experiments, i.e., the PD100 and PI100 impulse experiments described in Joos et.
al. (2013) where 100 Gt C is instantly added to preindustrial and near present
day simulated climates. However, our experiments also prescribe a negative CDR
pulse as opposed to just adding $CO_2$ to the atmosphere. Two experiments are
desirable because the Earth system response to $CO_2$ removal will be different
when starting from an equilibrium state versus starting from a perturbed state
(Zickfeld et al., 2016). One particular goal of these experiments is to estimate a
Global Cooling Potential (GCP) metric based on a CDR Impulse Response
Function ($IRF_{CDR}$). Such a metric will be useful for calculating how much $CO_2$ is
removed by DAC and how much DAC is needed to achieve a particular climate
target.
The third experiment, which focuses on "negative emissions", is based on
the Shared Socio-economic Pathway (SSP) 5-3.4-overshoot scenario and its long-
term extension (Kriegler et al., 2016; O'Neill et al., 2016). This scenario is of
interest to CDR-MIP because after an initially high level of emissions, which
follows the SSP5-8.5 unmitigated baseline scenario until 2040, $CO_2$ emissions are
rapidly reduced with net $CO_2$ emissions becoming negative after the year 2070
and continuing to be so until the year 2190 when they reach zero.  In the original
SSP5-3.4-OS scenario, the negative emissions are achieved using BECCS.
However, as stated earlier there is currently no practical way to design a good
multi-model BECCS experiment. Therefore, in our experiments negative
emissions are achieved by simply removing $CO_2$ from the atmosphere and
assuming that it is permanently stored in a geological reservoir. While this may
violate the economic assumptions underlying the scenario, it still provides an
opportunity to explore the response of the climate and carbon cycle to
potentially achievable levels of negative emissions.

According to calculations done with a simple climate model, MAGICC

version 6.8.01 BETA (Meinshausen et al., 2011; O'Neill et al., 2016), the SSP5-3.4-
OS scenario considerably overshoots the 3.4 W m$^{-2}$ forcing level, with a peak
global mean temperature of about 2.4° C, before returning to 3.4 W m$^{-2}$ at the
end of the century. Eventually in the long-term extension of this scenario, the
forcing stabilizes just above 2 W m$^{-2}$, with a global mean temperature that should
equilibrate at about 1.25° C above pre-industrial temperatures. Thus, in addition
to allowing an investigation into the response of the climate and carbon cycle to
negative emissions, this scenario also provides the opportunity to investigate
issues of reversibility, albeit on a shorter timescale and with less of an
"overshoot" than in experiment *CDR1-reversibility*.

**4.2.1 Instantaneous $CO_2$ removal / addition from an unperturbed climate**
**experimental protocol (*CDR2--pi-pulse*)**

This idealized Tier 1 experiment is designed to investigate how the Earth

system responds to DAC when perturbed from an equilibrium state (Fig. 3, Table
3). The idea is to provide a baseline system response that can later be compared
to the response of a perturbed system, i.e., experiment *CDR2--yr2010-pulse*
(Section 4.2.3). By also performing another simulation where the same amount
of $CO_2$ is added to the system, it will be possible to diagnose if the system
responds in an inverse manner when the $CO_2$ pulse is positive. Many modelling
groups will have already conducted the prerequisite simulation for this
experiment in preparation for other modelling research, e.g., during model spin-
up or for CMIP, which should minimize the effort needed to perform the
complete experiment. The protocol is as follows:

*Prerequisite simulation -* Control simulation under preindustrial conditions with
freely evolving $CO_2$. All boundary conditions (solar forcing, land use, etc.) are
expected to remain constant. This is also the CMIP5 *esmControl* simulation
(Taylor et al., 2012) and the CMIP6 *esm-piControl* simulation (Eyring et al.,
2016). Note that this is exactly the same as PI100 run 4 in Joos et. al. (2013).

*esm-pi-cdr-pulse* simulation - As in *esm-Control* or *esm-piControl*, but with 100 Gt
C instantaneously (within 1 time step) removed from the atmosphere in year 10.
If models have $CO_2$ spatially distributed throughout the atmosphere, we suggest
removing this amount in a uniform manner. After the negative pulse ESMs
should continue the run for at least 100 years, while EMICs and box models are
encouraged to continue the run for at least 1000 years (and up to 5000 years if
possible). Figure 4 shows example *esm-pi-cdr-pulse* model responses.

*esm-pi-~~co2pulse~~ CO2pulse* simulation - The same as *esm-pi-cdr-pulse*, but add a
positive 100 Gt C pulse (within 1 time step) as in Joos et. al. (2013), instead of a
negative one.  If models have $CO_2$ spatially distributed throughout the
atmosphere, we suggest adding $CO_2$ in a uniform manner. Note that this would be
exactly the same as the PI100 run 5 in Joos et. al. (2013) and can thus, be
compared to this earlier study.

**4.2.3 Instantaneous $CO_2$ removal from a perturbed climate experimental**
**protocol (*CDR~~2~~-* *yr2010-pulse*)**

This Tier 3 experiment is designed to investigate how the Earth system
responds when $CO_2$ is removed from an anthropogenically-altered climate not in
equilibrium (Fig. 5, Table 4). Many modelling groups will have already conducted
part of the first run of this experiment in preparation for other modelling
research, e.g., CMIP, and may be able to use a "restart" file to initialize the first
run, which should reduce the effort needed to perform the complete experiment.

*Prerequisite simulation* - Prescribed $CO_2$ run. Historical atmospheric $CO_2$ is
prescribed until a concentration of 389ppm is reached (~year 2010; Fig. 5 top
panel). Other historical forcing, i.e., from CMIP, should also be applied.  An
existing run or setup from CMIP5 or CMIP6 may also be used to reach a $CO_2$
concentration of 389ppm, e.g., the RCP 8.5 CMIP5 simulation or the CMIP6
*historical* experiment. During this run, compatible emissions should be
frequently diagnosed (at least annually).

~~*yr2010co2*~~ *yr2010CO2* simulation - Atmospheric $CO_2$ should be held constant at
389 ppm with other forcing, like land use and aerosol emissions, also held
constant (Fig. 5 top panel). ESMs should continue the run at 389ppm for at least
105 years, while EMICs and box models are encouraged to continue the run for
as long as needed for the subsequent simulations (e.g., 1000+ years). During this
run, compatible emissions should be frequently diagnosed (at least annually).
Note that when combined with the prerequisite simulation described above this
is exactly the same as the PD100 run 1 in Joos et. al. (2013).

*esm-* ~~*hist*~~ *yr2010~~co~~CO2-control* simulation - Diagnosed emissions control run. The
model is initialized from the pre-industrial period (i.e., using a restart from
either *piControl* or *esm-piControl*) with the emissions diagnosed in the *historical*
and ~~*yr2010co2*~~ *yr2010CO2* simulations, i.e., year 1850 to approximately year
2115 for ESMs and longer for EMICs and box models (up to 5000 years). All
other forcing should be as in the *historical* and ~~*yr2010co2*~~ *yr2010CO2*
simulations. Atmospheric $CO_2$ must be allowed to freely evolve. The results
should be quite close to those in the *historical* and ~~*yr2010co2*~~ *yr2010CO2*
simulations.  If there are significant differences, e.g., due to climate-carbon cycle
feedbacks that become evident when atmospheric $CO_2$ is allowed to freely
evolve, then they must be diagnosed and used to adjust the $CO_2$ emission forcing.
In some cases it may be necessary to perform an ensemble of simulations to
diagnose compatible emissions.  Note that this is exactly the same as the PD100
run 2 in Joos et. al. (2013). As in Joos et al. (2013), if computational time is an
issue and if a group is sure that $CO_2$ remains at a nearly constant value with the
emissions diagnosed in ~~*yr2010co2*~~*yr2010CO2*, the *esm-*~~*hist*~~*-*
*~~yr2010co2~~yr2010CO2-control* simulation may be skipped. This may only apply to
ESMs and it is strongly recommended to perform the *esm_~~-hist~~-*
*~~yr2010co2~~yr2010CO2-control* simulation to avoid model drift.

*esm-~~yr2010co2~~yr2010CO2-cdr-pulse* simulation - $CO_2$ removal simulation. Setup
is initially as in the *esm~~-hist~~-yr2010co2yr2010CO2-control* simulation. However, a
"negative" emissions pulse of 100 GtC is subtracted instantaneously (within 1
time step) from the atmosphere 5 years after the time at which $CO_2$ was held
constant in the *esm-~~hist-~~yr2010~~co~~CO2-control* simulation (this should be at the
beginning of the year 2015), with the run continuing thereafter for at least 100
years (up to 5000 years, if possible). If models have $CO_2$ spatially distributed
throughout the atmosphere, we suggest removing this amount in a uniform
manner. It is crucial that the negative pulse be subtracted from a constant
background concentration of ~389 ppm. All forcing, including $CO_2$ emissions,
must be exactly as in the *esm-~~hist--~~yr2010~~co~~CO2-control* simulation so that the
only difference between these runs is that this one has had $CO_2$ instantaneously
removed from the atmosphere.

*esm-~~yr2010co2~~yr2010CO2-noemit* - A zero $CO_2$ emissions control run. Setup is
initially as in the *esm-~~yr2010co2~~yr2010CO2-cdr-pulse* simulation. However, at the
time of the "negative" emissions pulse in the *esm-~~yr2010co2~~yr2010CO2-cdr-pulse*
simulation, emissions are set to zero with the run continuing thereafter for at
least 100 years. If possible extend the runs for at least 1000 years (and up to
5000 years). All other forcing must be exactly as in the *esm-*
*~~yr2010co2~~yr2010CO2-control* simulation. This experiment will be used to isolate
the Earth system response to the negative emissions pulse in the *esm-*
*~~yr2010co2~~yr2010CO2-cdr-pulse* simulation, which convolves the response to the
negative emissions pulse with the lagged response to the preceding positive $CO_2$
emissions (diagnosed with the zero emissions simulation). The response to the
negative emissions pulse will be calculated as the difference between *esm-*
*~~yr2010co2~~yr2010CO2-cdr-pulse* and *esm-~~yr2010co2~~yr2010CO2-noemit*
simulations.

*esm-~~yr2010co2~~yr2010CO2-~~co2pulse~~CO2pulse* simulation - $CO_2$ addition
simulation. Setup is initially as in the *esm-~~yr2010co2~~yr2010CO2-cdr-pulse*
simulation. However, a "positive" emissions pulse of 100 GtC is added
instantaneously (within 1 time step), with the run continuing thereafter for a
minimum of 100 years.  If models have $CO_2$ spatially distributed throughout the
atmosphere, we suggest adding $CO_2$ in a uniform manner. If possible extend the
runs for at least 1000 years (and up to 5000 years).  It is crucial that the positive
pulse be added to a constant background concentration of ~389 ppm.  All
forcing, including $CO_2$ emissions, must be exactly as in the *esm-~~hist~~-*
*~~yr2010co2~~yr2010CO2-control* simulation so that the only difference between
these runs is that this one has had $CO_2$ instantaneously added to the atmosphere.
Note that this would be exactly the same as PD100 run in Joos et. al. (2013). This
will be used to investigate if, after positive and negative pulses, carbon cycle and
climate feedback responses, which are expected to be opposite in sign, differ in
magnitude and temporal scale. The results can also be compared to Joos et. al.

(2013).


**4.2.5 Emission driven SSP5-3.4-OS experimental protocol (*~~CDR2~~-**
**-overshoot*)**

This Tier 2 experiment explores CDR in an "overshoot" climate change

scenario, the SSP5-3.4-OS scenario (Fig. 6, Table 5). To start, groups must
perform the CMIP6 emission driven historical simulation, *esm-hist*.  Then using
this as a starting point, conduct an emissions-driven SSP5-3.4-OS scenario
simulation, *esm-ssp534-over*, (starting on January 1, 2015) that includes the long-
term extension to the year 2300, *esm-ssp534-over-ext*. All non-$CO_2$ forcing should
be identical to that in the ScenarioMIP *ssp534-over* and *ssp534-over-ext*
simulations. If computational resources are sufficient, we recommend that the
*esm-ssp534-over-ext* simulation be continued for at least another 1000 years with
year 2300 forcing, i.e., the forcing is held constant at year 2300 levels as the
simulation continues for as long as possible; up to 5000 years, to better
understand processes that are slow to equilibrate, e.g., ocean carbon and heat
exchange or permafrost dynamics.

**4.3 Afforestation/reforestation experiment (*CDR3-afforestation*)**



Enhancing the terrestrial carbon sink by restoring or extending forest
cover, i.e., reforestation and afforestation, has often been suggested as a potential
CDR option (National Research Council, 2015; The Royal Society, 2009).
Enhancing this sink is appealing because terrestrial ecosystems have
cumulatively absorbed over a quarter of all fossil fuel emissions (Le Quéré et al.,
2016) and could potentially sequester much more. Most of the key questions
concerning land use change are being addressed by LUMIP (Lawrence et al.,
2016). These include investigations into the potential and side effects of
afforestation/reforestation to mitigate climate change, for which they have
designed four experiments (LUMIP Phase 2 experiments). However, three of
these experiments are $CO_2$ concentration driven, and thus are unable to fully
investigate the climate-carbon cycle feedbacks that are important for CDR-MIP.
The LUMIP experiment where $CO_2$ emissions force the simulation, *esm-ssp585-*
*ssp126Lu*, will allow for climate-carbon cycle feedbacks to be investigated.
Unfortunately, since this experiment ends in the year 2100 it is too short to
answer some of the key CDR-MIP questions (Section 1.2). We have therefore
decided to extend this LUMIP experiment within the CDR-MIP framework as a
Tier 2 experiment (Table 6) to better investigate the longer-term CDR potential
and risks of afforestation/reforestation.
The LUMIP experiment, *esm-ssp585-ssp126Lu,* simulates
afforestation/reforestation by combining a high SSP $CO_2$ emission scenario,
SSP5-8.5, with a future land use change scenario from an alternative SSP
scenario, SSP1-2.6, which has much greater afforestation/reforestation (Kriegler
et al., 2016; Lawrence et al., 2016). By comparing this combination to the SSP5-
8.5 baseline scenario, it will be possible to determine the CDR potential of this
particular afforestation/reforestation scenario in a high $CO_2$ world. This is
similar to the approach of Sonntag et al. (2016) using RCP 8.5 emissions
combined with prescribed RCP 4.5 land use.

**4.3.1 *CDR3-afforestation* Afforestation/reforestation experimental protocol**

*Prerequisite simulations* - Conduct the C4MIP emission-driven *esm-ssp585* simulation, which is a control run, and the LUMIP Phase 2 experiment *esm-ssp585-ssp126Lu* (Lawrence et al., 2016). Generate restart files in the year 2100.

*esm-ssp585-ssp126Lu-ext* simulation - Using the year 2100 restart from the *esm-ssp585-ssp126Lu* experiment, continue the run with the same LUMIP protocol (i.e., an emission driven SSP5-8.5 simulation with SSP1-2.6 land use instead of SSP5-8.5 land use) until the year 2300 using the SSP5-8.5 and SSP1-2.6 long-term extension data (O'Neill et al., 2016). If computational resources are sufficient, we recommend that the simulation be continued for at least another 1000 years with year 2300 forcing (i.e., forcing is held at year 2300 levels as the simulation continues for as long as possible; up to 5000 years). This is to better understand processes that are slow to equilibrate, e.g., ocean carbon and heat exchange or permafrost dynamics, and the issue of permanence.

*esm-ssp585ext* simulation - The emission-driven esmSSP5-8.5 simulation must be extended beyond the year 2100 to serve as a control run for the *esm-ssp585-ssp126Lu-ext* simulation. This will require using the ScenarioMIP *ssp585-ext* forcing, but driving the model with $CO_2$ emissions instead of prescribing the $CO_2$ concentration. If computational resources are sufficient, the simulation should be extended even further than in the official SSP scenario, which ends in year 2300, by keeping forcing constant after this time (i.e., forcing is held at year 2300 levels as the simulation continues for as long as possible; up to 5000 years).

## 4.4. Ocean alkalinization experiment (*CDR4-ocean-alk*)

Enhancing the natural process of weathering, which is one of the key negative climate-carbon cycle feedbacks that removes $CO_2$ from the atmosphere on long time scales (Colbourn et al., 2015; Walker et al., 1981), has been proposed as a potential CDR method (National Research Council, 2015; The Royal Society, 2009). Enhanced weathering ideas have been proposed for both the terrestrial environment (Hartmann et al., 2013) and the ocean (Köhler et al.,

2010; Schuiling and Krijgsman, 2006). We focus on the alkalinization of the
ocean given its capacity to take up vast quantities of carbon over relatively short
time periods and its potential to reduce the rate and impacts of ocean
acidification (Kroeker et al., 2013). The idea is to dissolve silicate or carbonate
minerals in seawater to increase total alkalinity. Total alkalinity, which can
chemically be defined as the excess of proton acceptors over proton donors with
respect to a certain zero level of protons, is a measurable quantity that is related
to the concentrations of species of the marine carbonate system (Wolf-Gladrow
et al., 2007). It plays a key role determining the air-sea gas exchange of $CO_2$
(Egleston et al., 2010). When total alkalinity is artificially increased in surface
waters, it basically allows more $CO_2$ to dissolve in the seawater and be stored as
ions such as bicarbonate or carbonate, i.e., the general methodology increases
the carbon storage capacity of seawater.
Theoretical work and idealized modelling studies have suggested that
ocean alkalinization may be an effective CDR method that is more limited by
logistic constraints (e.g., mining, transport, and mineral processing) rather than
natural ones, such as available ocean area, although chemical constraints and
side effects do exist (González and Ilyina, 2016; Ilyina et al., 2013; Keller et al.,
2014; Köhler et al., 2010, 2013). One general side effect of ocean alkalinization, is
that it increases the buffering capacity and pH of the seawater. While such a side
effect could be beneficial or even an intended effect to counter ocean
acidification (Feng et al., 2016), high levels of alkalinity may also be detrimental
to some organisms (Cripps et al., 2013). Ocean alkalinization likely also has
method specific side effects. Many of these side effects are related to the
composition of the alkalizing agent, e.g., olivine may contain nutrients or toxic
heavy metals, which could affect marine organisms and ecosystems (Hauck et al.,
2016; Köhler et al., 2013). Other side effects could be caused by the mining,
processing, and transport of the alkalizing agent, which in some cases may offset
the $CO_2$ sequestration potential of specific ocean alkalinization methods (e.g.,
through $CO_2$ release by fossil fuel use or during the calcination of $CaCO_3$)
(Kheshgi, 1995; Renforth et al., 2013).
Although previous modelling studies have suggested that ocean
alkalinization may be a viable CDR method, these studies are not comparable due
to different experimental designs. Here we propose an idealized Tier 2
experiment (Table 7) that is designed to investigate the response of the climate
system and carbon cycle to ocean alkalinization. The amount of any particular
alkalizing agent that could be mined, processed, transported, and delivered to
the ocean in a form that would easily dissolve and enhance alkalinity is poorly
constrained (Köhler et al., 2013; Renforth et al., 2013). Therefore, the amount of
alkalinity that is to be added in our experiment is set (based on exploratory
simulations conducted with the CSIRO-Mk3L-COAL model) to have a cumulative
effect on atmospheric $CO_2$ by the year 2100 that is comparable to the amount
removed in the CDR-MIP instantaneous DAC simulations, i.e., an atmospheric
reduction of ~100 Gt C; experiments *CDR2-pi-pulse* and *CDR2-yr2010-pulse*.
The idea here is not to test the maximum potential of such a method, which
would be difficult given the still relatively coarse resolution of many models and
the way in which ocean carbonate chemistry is simulated, but rather to compare
the response of models to a significant alkalinity perturbation. We have also
included an additional "termination" simulation that can be used to investigate
an abrupt stop in ocean alkalinization deployment.

**4.4.1 *CDR4-o Ocean-alkalinization* experimental protocol**

**Formatted:** Font: Italic
**Formatted:** Font: Italic
**Formatted:** Font: Italic


Prerequisite simulation - Conduct the C4MIP emission-driven *esm-ssp585*
simulation as described by Jones et al., (2016b). This is the SSP5-8.5 high $CO_2$
emission scenario, and it serves as the control run and branching point for the
ocean alkalinization experiment.  A restart must be generated at the end of the
year 2019.

*esm-ssp585-ocean-alk* simulation - Begin an 80 year run using the *esm-ssp585*
year 2020 restart (starting on Jan. 1, 2020) and add 0.14 Pmol Total Alkalinity
(TA) yr$^{-1}$ to the upper grid boxes of each model's ocean component, i.e., branch
from the C4MIP *esm-ssp585* simulation in 2020 until 2100. The alkalinity
additions should be limited to mostly ice free, year-round ship accessible waters,
which for simplicity should set to be between 70°N and 60°S (note that this
ignores the presence of seasonal sea-ice in some small regions). For many
models, this will in practice result in an artificial TA flux at the air-sea interface
with realized units that might, for example, be something like $\mu mol\ TA\ s^{-1}\ cm^{-2}$.
Adding 0. 14 Pmol TA $yr^{-1}$ is equivalent to adding 5.19 Pg $yr^{-1}$ of an alkalizing
agent like $Ca(OH)_2$ or 4.92 Pg $yr^{-1}$ of forsterite ($Mg_2SiO_4$), a form of olivine
[assuming theoretical net instant dissolution reactions which for every mole of
$Ca(OH)_2$ or $Mg_2SiO_4$ added sequesters 2 or 4 moles, respectively, of $CO_2$ (Ilyina et
al., 2013; Köhler et al., 2013)]. As not all models include marine iron or silicate
cycles, the addition of these nutrients, which could occur if some form of olivine
were used as the alkalizing agent, is not considered here. All other forcing is as in
the *esm-ssp585* control simulation. If the ocean alkalinization termination
simulation (below) is to be conducted, generate a restart at the beginning of the
year 2070.

Optional (Tier 3) *esm-ssp585-ocean-alk-stop* simulation - Use the year 2070
restart from the *esm-ssp585-ocean-alk* simulation and start a simulation
(beginning on Jan. 1, 2070) with the SPP5-8.5 forcing, but without adding any
additional alkalinity. Continue this run until the year 2100, or beyond, if
conducting the *esm-ssp585-ocean-alk-ext* simulation (below).

Optional (Tier 3) ocean alkalinization extension simulations:

*esm-ssp585ext* simulation - If groups desire to extend the ocean alkalinization
experiment beyond the year 2100, an optional simulation may be conducted to
extend the control run using forcing data from the ScenarioMIP *ssp585ext*
simulation, i.e., conduct a longer emission-driven control run, *esm-ssp585ext.*
This extension is also a control run for those conducting the CDR-MIP *CDR3* 
*afforestation/reforestation* simulation (Section 4.3). If computational resources
are sufficient, the simulation should be extended even further than in the official
SSP scenario, which ends in year 2300, by keeping the forcing constant after this
time (i.e., forcing is held at year 2300 levels as the simulation continues for as
long as possible; up to 5000 years).

*esm-ssp585-ocean-alk-ext* simulation - Continue the ocean alkalinization
experiment described above (i.e., adding 0.14 Pmol Total Alkalinity (TA) $yr^{-1}$ to
the upper grid boxes of each model's ocean component) beyond the year 2100
(up to 5000 years) using forcing from the *esm-ssp585-ext* simulation.

**5. Model output, data availability, and data use policy**
**5.1 Gridded model output**

Models capable of generating gridded data must use a NetCDF format. The
output (see Appendix A web link for the list of requested variables) follows the
CMIP6 output requirements in frequency and structure. This allows groups to
use CMOR software (Climate Model Rewriter Software, available at
http://cmor.llnl.gov/) to generate the files that will be available for public
download (Section 5.5). ~~CMOR3 tables for CDR-MIP are available at www.kiel-~~
~~earth-institute.de/files/media/downloads/CDRmon.json (table for monthly~~
~~output) and www.kiel-earth-institute.de/files/media/downloads/CDRga.json~~
~~(table for global annual mean output).~~ The resolution of the data should be as
close to native resolution as possible, but on a regular grid. Please note as
different models have different formulations, only applicable outputs need be
provided. However, groups are encouraged to generate additional output, i.e.,
whatever their standard output variables are, and can also make this data
available (preferably following the CMIP6 CMOR standardized naming
structure).

**5.2 Conversion factor Gt C to ppm**

For experiments where carbon must be converted between GtC (or Pg)
and ppm $CO_2$, please use a conversion factor of 2.12 GtC per ppm $CO_2$ to be
consistent with Global Carbon Budget (Le Quere et al., 2015) conversion factors.

**5.3 Box model output**

For models that are incapable of producing gridded NetCDF data (i.e., box
models), output is expected to be in an ASCII format (Appendix B). All ASCII files
are expected to contain tabulated values (at a minimum global mean values),
with at least two significant digits for each run. Models must be able to calculate
key carbon cycle variables (Appendix C) to participate in CDR-MIP experiments
~~C1~~ *CDR-reversibility*, ~~and C2~~ *CDR-pi-pulse*, and *CDR-yr2010-pulse*. Please submit
these files directly to the corresponding author who will make them available for
registered users to download from the CDR-MIP website.

**5.4 Model output frequency**

The model output frequency is listed in Table 8. In all experiments box
models and EMICs without seasonality are expected to generate annual mean
output for the duration of the experiment, while models with seasonality are
expected to generate higher spatial resolution data, i.e., monthly, for most
simulations.
In experiment *CDR~~1~~-reversibility* for the control run, *piControl*, we request
that 100 years of 3-D model output be written monthly (this should be the last
100 years if conducting a 500+ year run for CMIP6). For the *1pctCO2* and
*1pctCO2-cdr* simulations 3-D model output should also be written monthly, i.e.,
as the atmospheric $CO_2$ concentration is changing. We suggest that groups that
have already performed the *piControl* and *1pctCO2* simulations for CMIP5 or
CMIP6 with an even higher output resolution (e.g., daily) continue to use this
resolution for the *1pctCO2-cdr* simulation, as this will facilitate the analysis. For
groups continuing the simulations for up to 5000 years after $CO_2$ has returned to
284.7 ppm, at a minimum, annual global mean values (non-gridded output)
should be generated after the initial minimum 60 years of higher resolution
output.
For experiment *CDR~~2~~-pi-pulse* if possible, 3-D model output should be
written monthly for 10 years before the negative pulse and for 100 years
following the pulse. For groups that can perform longer simulations, e.g.,
thousands of years, at a minimum, annual global mean values (non-gridded
output) should be generated. Data for the control run, i.e., the equilibrium
simulation *esm-piControl*, must also be available for analytical purposes. CMIP
participants may provide a link to the *esm-Control* or *esm-piControl* data on the
ESGF.
For experiment *CDR2--yr2010-pulse* the *historical* and ~~*yr2010co2*~~
*yr2010CO2* simulations output is only needed to diagnose annual $CO_2$ emissions
and will not be archived on the ESGF. Gridded 3-D monthly mean output for the
*esm-~~hist-yr2010co2~~yr2010CO2-control* (starting in the year 2010)*, esm-*
*~~yr2010co2~~yr2010CO2-cdr-pulse, esm-~~yr2010co2~~yr2010CO2-noemit,* and *esm-*
*~~yr2010co2~~yr2010CO2-~~co2pulse~~CO2pulse* simulations should be written for the
initial 100 years of the simulation. Thereafter, for groups that can perform longer
simulations (up to 5000 years), at a minimum annual global mean values (non-
gridded output) should be generated. CMIP participants are requested to provide
a link to the *historical* simulation data on the ESGF.
For experiment *CDR~~2~~--overshoot,* if possible, 3-D model output should be
written monthly until the year 2300. We suggest that groups that have already
performed the ScenarioMIP *ssp534-over* and *ssp534-over-ext* and C4MIP *ssp534-*
*over-bgc* and *ssp534-over-bgcExt* CMIP6 simulations with an even higher output
resolution (e.g., daily) continue to use this resolution as this will facilitate
analyses. For groups that can perform longer simulations, e.g., thousands of
years, at a minimum annual global mean values (non-gridded output) should be
generated for every year beyond 2300. We recommend that CMIP participants
provide a link to the *esm-hist* data on the ESGF. For analytical purposes, we also
request that ScenarioMIP and C4MIP participants provide links to any completed
*ssp534-over, ssp534-over-ext*, *ssp534-over-bgc* and *ssp534-over-bgcExt* simulation
data on the ESGF.
For experiment *CDR~~3~~-afforestation* if possible, 3-D model output should
be written monthly until the year 2300. LUMIP participants may provide a link to
the *esm-hist* and *esm-ssp585-ssp126Lu* data on the ESGF for the first portions of
this run (until the year 2100). For groups that can perform longer simulations,
e.g., thousands of years, at a minimum annual global mean values (non-gridded
output) should be generated for every year beyond 2300.
For experiment *CDR~~4~~-ocean-alk* if possible, 3-D gridded model output
should be written monthly for all simulations. For groups that can perform
longer simulations, e.g., thousands of years, at a minimum annual global mean
values (non-gridded output) should be generated for every year beyond 2300.

**5.5 Data availability and use policy**

The model output from the CDR-MIP experiments described in this paper
will be publically available. All gridded model output will, to the extent possible,
be distributed through the Earth System Grid Federation (ESGF). Box model
output will be available via the CDR-MIP website (http://www.kiel-earth-
institute.de/cdr-mip-data.html). The CDR-MIP policy for data use is that if you
use output from a particular model, you should contact the modeling group and
offer them the opportunity to contribute as authors. Modeling groups will
possess detailed understanding of their models and the intricacies of performing
the CDR-MIP experiments, so their perspectives will undoubtedly be useful. At
minimum, if the offer of author contribution is not taken up, CDR-MIP and the
model groups should be credited in acknowledgments with for example a
statement like: "*We acknowledge the Carbon Dioxide Removal Model*
*Intercomparison Project leaders and steering committee who are responsible for*
*CDR-MIP and we thank the climate modelling groups (listed in Table XX of this*
*paper) for producing and making their model output available.*"
The natural and anthropogenic forcing data that are required for some
simulations are described in several papers in the Geoscientific Model
Development CMIP6 special issue. These data will be available on the ESGF.
Links to all forcing data can also be found on the CMIP6 Panel website
(https://www.wcrp-climate.org/wgcm-cmip/wgcm-cmip6). CMIP6 and CMIP5
data should be acknowledged in the standard way.

**6. CDR-MIP outlook and conclusion**

It is anticipated that this will be the first stage of an ongoing project
exploring CDR. CDR-MIP welcomes input on the development of other (future)
experiments and scenarios. Potential future experiments could include Biomass
Energy with Carbon Capture and Storage (BECCS) or ocean fertilization. Future
experiments could also include the removal of non-$CO_2$ greenhouse gases, e.g.,
methane, as these in many cases have a much higher global warming potential
(de_Richter et al., 2017; Ming et al., 2016). We also envision that it will be
necessary to investigate the simultaneous deployment of several CDR or other
greenhouse gas removal methods since early studies suggest that there is likely
not an individually capable method (Keller et al., 2014). It is also anticipated that
scenarios will be developed that might combine Solar Radiation Management
(SRM) and CDR in the future, such as a joint GeoMIP (Geoengineering Model
Intercomparison Project) CDR-MIP experiment.
In addition to reductions in anthropogenic $CO_2$ emissions, it is very likely
that CDR will be needed to achieve the climate change mitigation goals laid out in
the Paris Agreement. The potential and risks of large scale CDR are poorly
quantified, raising important questions about the extent to which large scale CDR
can be depended upon to meet Paris Agreement goals. ~~This project~~As an
endorsed CMIP6 activity, CDR-MIP, is designed to help us better understand how
the Earth system might respond to CDR. Over the past two years the CDR-MIP
team has developed a set of numerical experiments to be performed with Earth
system models of varying complexity. The aim of these experiments is to provide
coordinated simulations and analyses that addresses several key CDR
uncertainties including:

• The degree to which CDR could help mitigate climate change or even
reverse it.

• The potential effectiveness and risks/benefits of different CDR proposals
with a focus on direct $CO_2$ air capture, afforestation/reforestation, and
ocean alkalinization.

• To inform how CDR might be appropriately accounted for within an Earth
system framework and during scenario development.

We anticipate that there will be numerous forthcoming studies that utilize
CDR-MIP data. The model output from the CDR-MIP experiments will be
publically available and we welcome and encourage interested parties to
download this data and utilize it to further investigate CDR.

**7. Code and/or data availability**

As described in Section 5.5, the output from models participating in CDR-
MIP will be made publically available.  This will include data used in exemplary
Figs. 2 and 4.  All gridded model output will be distributed through the Earth
System Grid Federation (ESGF) with digital object identifiers (DOIs) assigned.
Box model output will be available via the CDR-MIP website (http://www.kiel-
earth-institute.de/cdr-mip-data.html).  The code from the models used to
generate the exemplary figures in this document (Figs. 2 and 4, Appendix D) will
be made available here via a web link when this manuscript is accepted for
publication.   To obtain code from modelling groups who are participating in
CDR-MIP please contact the modelling group using the contact information that
accompanies their data.

*Acknowledgements.*  D. P. Keller and N. Bauer acknowledge funding received from
the German Research Foundation's Priority Program 1689 "Climate Engineering"
(project CDR-MIA; KE 2149/2-1). K. Zickfeld acknowledges support from the
Natural Sciences and Engineering Research Council of Canada (NSERC)
Discovery grant program.  The Pacific Northwest National Laboratory is
operated for the U.S. Department of Energy by Battelle Memorial Institute under
contract DE-AC05-76RL01830. D. Ji acknowledges support from the National
Basic Research Program of China under grant number 2015CB953600. CDJ was
supported by the Joint UK BEIS/ Defra Met Office Hadley Centre Climate
Programme ( GA01101) and by the European Union' s Horizon 2020 research
and innovation programme under grant agreement No 641816 ( CRESCENDO).
H. Muri was supported by Norwegian Research Council grant 261862/E10.

**Appendix A.  Requested model output variables**

A spreadsheet of the requested model output variables and their format can be
found at: www.kiel-earth-institute.de/files/media/downloads/CDR-
MIP_model_output_requirements.pdf. Please note as different models have
different formulations, only applicable outputs need be provided. However,
groups are encouraged to generate additional output, i.e., whatever their
standard output variables are, and can also make this data available.

**Appendix B. Box model output formatting**

Box model ASCII formatting example:

File name format: RUNNAME_MODELNAME_Modelversion.dat
C1_MYBOXMODEL_V1.0_.dat
Headers and formats:
*Example:*
- Start each header comment line with a #
- *Line 1:* Indicate run name, e.g., "# *esm-pi-cdr-pulse* "
- *Line 2:* Provide contact address, e.g., "# B. Box, Uni of Box Models, CO2
Str.,   BoxCity 110110, BoxCountry"
- *Line 3:* Provide a contact email address, e.g., "# bbox@unibox.bx"
- *Line 4:* Indicate model name, version, e.g., "# MyBoxModel Version 2.2"
- *Line 5:* Concisely indicate main components, e.g., "# two ocean boxes
(upper and lower), terrestrial biosphere, and one atmospheric box"
- *Line 6:* Indicate climate sensitivity of model, the abbreviation TCS may be
used for transient climate sensitivity and ECS for equilibrium climate
sensitivity, e.g., "#   TCS=3.2 [deg C], ECS=8.1 [deg C]"
- *Line 7:* Description of non-$CO_2$ forcing applied, e.g., "# Forcing: solar"
- *Line 8:* Indicate the output frequency and averaging, e.g., "# Output: global
mean values"
- *Line 9:* List tabulated output column headers with their units in brackets
(see table below), e.g., "# year tas[K]"

Complete Header Example:
# *esm-pi-cdr-pulse*
# B. Box, Uni. of Box Models, CO2 Str., BoxCity 110110, BoxCountry
# bbox@unibox.bx
# MyBoxModel Version 2.2
# two ocean boxes (upper and lower), terrestrial biosphere, and one
atmospheric box
# TCS=3.2 deg C, ECS=8.1 deg C
# Forcing: solar
# Output: global mean values
# year tas[K] co2[Gt C] nep[Gt C yr-1] fgco2[Gt C yr-1]

**Appendix C. Requested box model output variables**

Table of requested box model output (at a minimum as global mean values). To
participate in CDR-MIP at a minimum the variables *tas, xco2,* and *fgco2* must be
provided.

| Long name | Column Header Name[*] | Units | Comments |
|---|---|---|---|
| Relative year | year | year | |
| Near-surface Air Temperature | tas | K | |
| Atmospheric $CO_2$ | xco2 | ppm | |
| Surface Downward $CO_2$ flux into the ocean | fgco2 | kg m$^{-2}$ | This is the net air-to-ocean carbon flux (positive flux is into the ocean) |
| Total Atmospheric Mass of $CO_2$ | co2mass | kg | |
| Net Carbon Mass Flux out of Atmosphere due to Net Ecosystem Productivity on Land. | nep | kg m$^{-2}$ | This is the net air-to-land carbon flux (positive flux is into the land) |
| Total ocean carbon | cOcean | Gt C | If the ocean contains multiple boxes this output can also be provided, e.g., as cOcean_up and cOcean_low for upper and lower ocean boxes |

| | | | |
|---|---|---|---|
| Total land carbon | cLand | Gt C | This is the sum of all C pools |
| Ocean Potential Temperature | thetao | K | Please report a mean value if there are multiple ocean boxes |
| Upper ocean pH | pH | 1 | Negative log of hydrogen ion concentration with the concentration expressed as mol H kg$^{-1}$. |
| Carbon Mass Flux out of Atmosphere due to Net Primary Production on Land | npp | kg m$^{-2}$ | This is calculated as gross primary production – autotrophic respiration (gpp-ra) |
| Carbon Mass Flux into Atmosphere due to Heterotrophic Respiration on Land | rh | kg m$^{-2}$ | |
| Ocean Net Primary Production by Phytoplankton | intpp | kg m$^{-2}$ | |
| | | | |


*Column header names follow the CMIP CMOR notation when possible

**Appendix D. Model descriptions**

The two models used to develop and test CDR–MIP experimental

protocols and provide example results (Figs. 2 and 4) are described below.

The University of Victoria Earth System Climate model (UVic), version 2.9

consists of three dynamically coupled components: a three-dimensional general
circulation model of the ocean that includes a dynamic-thermodynamic sea ice
model, a terrestrial model, and a simple one-layer atmospheric energy-moisture
balance model (Eby et al., 2013). All components have a common horizontal
resolution of 3.6° longitude x 1.8° latitude. The oceanic component, which is in
the configuration described by Keller et al. (2012), has 19 levels in the vertical
with thicknesses ranging from 50 m near the surface to 500 m in the deep ocean.
The terrestrial model of vegetation and carbon cycles (Meissner et al., 2003) is
based on the Hadley Center model TRIFFID (Top-down Representation of
Interactive Foliage and Flora Including Dynamics). The atmospheric energy-
moisture balance model interactively calculates heat and water fluxes to the

ocean, land, and sea ice. Wind velocities, which are used to calculate the momentum transfer to the ocean and sea ice model, surface heat and water fluxes, and the advection of water vapor in the atmosphere, are determined by adding wind and wind stress anomalies. These are determined from surface pressure anomalies that are calculated from deviations in pre-industrial surface air temperature to prescribed NCAR/NCEP monthly climatological wind data (Weaver et al., 2001). The model has been extensively used in climate change studies and is also well validated under pre-industrial to present day conditions (Eby et al., 2009, 2013; Keller et al., 2012).

The CSIRO-Mk3L-COAL Earth system model consists of a climate model, Mk3L (Phipps et al., 2011), coupled to a biogeochemical model of carbon, nitrogen and phosphorus cycles on land (CASA-CNP) in the Australian community land surface model, CABLE (Mao et al., 2011; Wang et al., 2010), and an ocean biogeochemical cycle model (Duteil et al., 2012; Matear and Hirst, 2003). The atmospheric model has a horizontal resolution of 5.6° longitude by 3.2° latitude, and 18 vertical layers. The land carbon model has the same horizontal resolution as the atmosphere. The ocean model has a resolution of 2.8° longitude by 1.6° latitude, and 21 vertical levels. Mk3L simulates the historical climate well, as compared to the models used for earlier IPCC assessments (Phipps et al., 2011). Furthermore, the simulated response of the land carbon cycle to increasing atmospheric $CO_2$ and warming are consistent with those from the Coupled Model Intercomparison Project Phase 5 (CMIP5) (Zhang et al., 2014). The ocean biogeochemical model was also shown to realistically simulate the global ocean carbon cycle (Duteil et al., 2012; Matear and Lenton, 2014).

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

**CDRMIP~~CDR-MIP~~ GMDD manuscript tables**

Table 1. Overview of CDRMIP experiments.  Note that each experiment is comprised of several individually named simulations (Tables 2-7).~~CDR-MIP experiments.~~  In the "Forcing methods" column, "All" means "all anthropogenic, solar, and volcanic forcing".   Anthropogenic forcing includes aerosol emissions, non-$CO_2$ greenhouse gas emissions, and land use changes.

| Short Name | Long Name | Tier | Experiment Description | Forcing methods | Major purpose |
|---|---|---|---|---|---|
| CDR-reversibility~~C1~~ | Climate and carbon cycle reversibility experiment | 1 | $CO_2$ prescribed to increase at 1% $yr^{-1}$ to 4x pre-industrial $CO_2$ and then decrease at 1% $yr^{-1}$ until again at a pre-industrial level, after which the simulation continues for as long as possible | $CO_2$ concentration prescribed | Evaluate climate reversibility |
| CDR-~~C2-~~pi-pulse | Instantaneous $CO_2$ removal / addition from an unperturbed climate experiment | 1 | 100 Gt C is instantly removed (negative pulse) from a steady-state pre-industrial atmosphere; 100 Gt C is instantly added (positive pulse) to a steady-state pre-industrial atmosphere | $CO_2$ concentration calculated (i.e., freely evolving) | Evaluate climate and C-cycle response of an unperturbed system to atmospheric $CO_2$ removal; comparison with the positive pulse response |
| CDR-~~C2-~~yr2010-pulse | Instantaneous $CO_2$ removal / addition from a perturbed climate experiment | 3 | 100 Gt C is instantly removed (negative pulse) from a near present-day atmosphere; 100 Gt C is instantly added (positive pulse) to a near present-day atmosphere | All; $CO_2$ concentration calculated (i.e., emission driven)* | Evaluate climate and C-cycle response of a perturbed system to atmospheric $CO_2$ removal; comparison with the positive pulse response |
| CDR-~~C2-~~overshoot | Emission driven SSP5-3.4-OS scenario experiment | 2 | SSP5-3.4-overshoot scenario where $CO_2$ emissions are initially high and then rapidly reduced, becoming negative | All; $CO_2$ concentration calculated (i.e., emission driven) | Evaluate the Earth system response to CDR in an overshoot climate change scenario |
| CDR-afforestation~~C3~~ | Afforestation/ reforestation experiment | 2 | Long-term extension of an experiment with forcing from a high $CO_2$ emission scenario (SSP5-8.5), but with land use prescribed from a scenario with high levels of afforestation and reforestation (SSP1-2.6) | All; $CO_2$ concentration calculated (i.e., emission driven) | Evaluate the long-term Earth system response to afforestation/ reforestation during a high $CO_2$ emission climate change scenario |
| CDR-ocean-alk~~C4~~ | Ocean alkalinization in a high $CO_2$ world experiment | 2 | A high $CO_2$ emission scenario (SSP5-8.5) with 0.14 Pmol $yr^{-1}$ alkalinity added to ice-free ocean surface waters from the year 2020 onward | All; $CO_2$ concentration calculated (i.e., emission driven) | Evaluate the Earth system response to ocean alkalinization during a high $CO_2$ emission climate change scenario |

*In this experiment $CO_2$ is first prescribed to diagnose emissions, however, the key simulations calculate the $CO_2$ concentration.

Table 2. Climate and carbon cycle reversibility experiment (*CDR-reversibilityC1*) simulations. All simulations are required to complete the experiment.

| CMIP6 Experiment Simulation ID | Simulation description | Owning MIP | Run length (years) | Initialized using a restart from |
|---|---|---|---|---|
| *piControl* | Pre-industrial prescribed $CO_2$ control simulation | CMIP6 DECK | 100* | The model spin-up |
| *1pctCO2* | Prescribed 1% $yr^{-1}$ $CO_2$ increase to 4× the pre-industrial level | CMIP6 DECK | 140** | *piControl* |
| *1pctCO2-cdr* | 1% $yr^{-1}$ $CO_2$ decrease from 4× the pre-industrial level until the pre-industrial $CO_2$ level is reached and held for as long as possible | CDRMIPC DR-MIP | 200 min. 5000 max. | *1pctCO2* |

*This CMIP6 DECK should have been run for at least 500 years. Only the last 100 years are needed as a control for *CDR-reversibilityC1*.
**This CMIP6 DECK experiment is 150 years long. A restart for *CDR-reversibilityC1* should be generated after 139 years when $CO_2$ is 4 times that of *piControl.*

Table 3. Instantaneous $CO_2$ removal from an unperturbed climate experiment (*CDR-C2 pi-pulse*) simulations. All simulations are required to complete the experiment.

| CMIP6 ExperimentSimulation ID | Simulation description | Owning MIP | Run length (years) | Initialized using a restart from |
|---|---|---|---|---|
| *esm-piControl* | Pre-industrial freely evolving $CO_2$ control simulation | CMIP6 DECK | 100* | The model spin up |
| *esm-pi-cdr-pulse* | 100 Gt C is instantly removed (negative pulse) from a pre-industrial atmosphere | CDRMIPC DR-MIP | 100 min. 5000 max. | *esm-piControl* |
| *esm-pi-CO2pulseco2pulse* | 100 Gt C is instantly added to (positive pulse) a pre-industrial atmosphere | CDRMIPC DR-MIP | 100 min. 5000 max. | *esm-piControl* |

*This CMIP6 DECK should have been run for at least 500 years. Only the last 100 years are needed as a control for *CDR-pi-pulseC2.1*.

Table 4. Instantaneous CO$_2$ removal from a perturbed climate experiment (*CDR-C2 yr2010-pulse*) simulations. All simulations are required to complete the experiment.

| CMIP6 ExperimentSimulation ID | Simulation description | Owning MIP | Run length (years) | Initialized using a restart from |
|---|---|---|---|---|
| *historical* | Historical atmospheric CO$_2$ (and other forcing) is prescribed until a concentration of 389ppm CO$_2$ is reached | CMIP6 DECK | 160* | *piControl* |
| *yr2010CO2yr2010co2* | Branching from *historical,* atmospheric CO$_2$ is held constant (prescribed) at 389ppm; other forcing is also held constant at the 2010 level | CDRMIPC DR-MIP | 105 min. 5000 max. | *historical* |
| *esm-yr2010CO2hist-yr2010co2-control* | Control run forced using CO$_2$ emissions diagnosed from *historical* and *yr2010co2* simulations; other forcing as in *historical* until 2010 after which it is constant | CDRMIPC DR-MIP | 265 min. 5160 max. | *esm-piControl* or *piControl* |
| *esm-yr2010CO2yr2010co2-noemit* | Control run that branches from *esm-hist-yr2010co2-control* in year 2010 with CO$_2$ emissions set to zero 5 years after the start of the simulation | CDRMIPC DR-MIP | 105 min. 5000 max. | *esm-yr2010CO2hist-yr2010co2-control* |
| *esm-yr2010CO2yr2010co2-cdr-pulse* | Branches from *esm-hist-yr2010co2-control* in year 2010 with 100 Gt C instantly removed (negative pulse) from the atmosphere 5 years after the start of the simulation | CDRMIPC DR-MIP | 105 min. 5000 max. | *esm-yr2010CO2hist-yr2010co2-control* |
| *esm-yr2010CO2-CO2pulseyr2010co2-co2pulse* | Branches from *esm-hist-yr2010co2-control* in year 2010 with 100 Gt C instantly added to (positive pulse) the atmosphere 5 years after the start of the simulation | CDRMIPC DR-MIP | 105 min. 5000 max. | *esm-yr2010CO2hist-yr2010co2-control* |

*This CMIP6 DECK continues until the year 2015 but only the first 160 years are need for *CDR-C2 yr2010-pulse*.

Table 5. Emission driven SSP5-3.5-OS scenario experiment (*CDR-C2-overshoot*) simulations. All simulations are required to complete the experiment.

| CMIP6 Experiment~~Simulation~~ ID | Simulation description | Owning MIP | Run length (years) | Initialized using a restart from |
|---|---|---|---|---|
| *esm-hist* | Historical simulation forced with $CO_2$ emissions | CMIP6 DECK | 265 | *esm-piControl* or *piControl* |
| *esm-ssp534-over* | $CO_2$ emission-driven SSP5-3.4 overshoot scenario simulation | CDRMIP~~C DR  MIP~~ | 85 | *esm-hist* |
| *esm-ssp534-over-ext* | Long-term extension of the $CO_2$ emission-driven SSP5-3.4 overshoot scenario | CDRMIP~~C DR  MIP~~ | 200 min. 5000 max. | *esm-ssp534-over* |

Table 6. Afforestation/ reforestation experiment (*CDR-afforestationC3*) simulations.  All simulations are required to complete the experiment.

| CMIP6 Experiment~~Simulation~~ ID | Simulation description | Owning MIP | Run length (years) | Initialized using a restart from |
|---|---|---|---|---|
| *esm-ssp585* | $CO_2$ emission driven SSP5-8.5 scenario | C4MIP | 85 | *esm-hist* |
| *esm-ssp585-ssp126Lu* | $CO_2$ emission driven SSP5-8.5 scenario with SSP1-2.6 land use forcing | LUMIP | 85 | *esm-hist* |
| *esm-ssp585-ssp126Lu-ext* | Long-term extension of the *esm-ssp585-ssp126Lu*~~CO₂ emission-driven SSP5-3.4 overshoot scenario~~ simulation | CDRMIP~~C DR  MIP~~ | 200 min. 5000 max. | *esm-ssp585-ssp126Lu* |
| *esm-ssp585ext* | Long-term extension of the $CO_2$ emission-driven SSP5-8.5 scenario | CDRMIP~~C DR  MIP~~ | 200 min. 5000 max. | *esm-ssp585* |

Table 7. Ocean alkalinization (_CDR-ocean-alk_~~C4~~) experiment simulations.  "Pr" in the Tier column indicates a prerequisite experiment.

| CMIP6 Experiment~~Simulation~~ ID | Tier | Simulation description | Owning MIP | Run length (years) | Initialized using a restart from |
|---|---|---|---|---|---|
| _esm-ssp585_ | Pr | $CO_2$ emission driven SSP5-8.5 scenario | C4MIP | 85 | _esm-hist_ |
| _esm-ssp585-ocn~~ocean~~-alk_ | 2 | SSP5-8.5 scenario with 0.14 Pmol $yr^{-1}$ alkalinity added to ice-free ocean surface waters from the year 2020 onward | CDRMIP~~C DR-MIP~~ | 65 | _esm-ssp585_ |
| _esm-ssp585-ocn~~ocean~~-alk-stop_ | 3 | Termination simulation to investigate an abrupt stop in ocean alkalinization in the year 2070 | CDRMIP~~C DR-MIP~~ | 30* | _esm-ssp585-ocn~~ocean~~-alk_ |
| _esm-ssp585ext_ | 3 | Long-term extension of the $CO_2$ emission-driven SSP5-8.5 scenario | CDRMIP~~C DR-MIP~~ | 200 min. 5000 max. | _esm-ssp585_ |
| _esm-ssp585-ocn~~ocean~~-alk-ext_ | 3 | Long-term extension of the _esm-ssp585-ocn~~ocean~~-alk_ simulation | CDRMIP~~C DR-MIP~~ | 200 min. 5000 max. | _esm-ssp585-ocn~~ocean~~-alk_ |

*If the _esm-ssp585ext_ simulation is being conducted this may be extended for more than 200 more years (up to 5000 years).

Table 8. Model output frequency for 3-D models with seasonality. Box models and EMICs without seasonality are expected to generate annual global mean output for the duration of all experiments.  For longer simulations (right column) if possible 3-D monthly data should be written out for one year every 100 years. For models with interannual variability, e.g., ESMs, monthly data should be written out for a 10-year period every 100 years so that a climatology may be developed. The years referred to in the table indicate simulations years, e.g. years from the start of the run, not that of any particular scenario.

| CDRMIP Experiment Short Name | Individual simulation output frequency | |
|---|---|---|
| | **Monthly gridded 3-D output** | **Annual global mean output + climatological output at 100 year intervals** |
| _CDR-reversibilityC1_ | _piControl_ (last 100 years)<br>_1pctCO2_<br>_1pctCO2-cdr_ (initial 200 years) | _1pctCO2-cdr_ (from year 200 onward) |
| _CDR-C2-pi-pulse_ | _esm-piControl_<br>_esm-pi-cdr-pulse_ (initial 100 years)<br>_esm-pi-CO2pulseco2pulse_ (initial 100 years) | _esm-pi-cdr-pulse_ (from year 100 onward)<br>_esm-pi-CO2pulseco2pulse_ (from year 100 onward) |
| _CDR-C2-yr2010-pulse_ | _esm-yr2010CO2hist-yr2010co2-control_ (initial 105 years)<br>_esm-yr2010CO2yr2010co2-noemit_<br>_esm-yr2010CO2yr2010co2-cdr-pulse_<br>_esm-yr2010CO2-CO2pulseyr2010co2-co2pulse_ | _esm- yr2010CO2hist-yr2010co2-control_<br>_esm-yr2010CO2yr2010co2-noemit_<br>_esm-yr2010CO2yr2010co2-cdr-pulse_<br>_esm-yr2010CO2-CO2pulseyr2010co2-co2pulse_ |
| _CDR-C2-overshoot_ | _esm-hist_<br>_esm-ssp534-over_<br>_esm-ssp534-over-ext_ (initial 200 years) | _esm-ssp534-over-ext_ (from year 200 onward)** |
| _CDR-afforestationC3_ | _esm-ssp585ext_  (initial 200 years)<br>_esm-ssp585-ssp126Lu_<br>_esm-ssp585-ssp126Lu-ext_  (initial 200 years) | _esm-ssp585ext_ (from year 200 onward)**<br>_esm-ssp585-ssp126Lu-ext_ (from year 200 onward)** |
| _CDR-ocean-alkC4_ | _esm-ssp585_<br>_esm-ssp585-ocnocean-alk_<br>_esm-ssp585-ocnocean-alk-stop_  (initial 200 years)<br>_esm-ssp585ext_  (initial 200 years)<br>_esm-ssp585-ocnocean-alk-ext_  (initial 200 years) | _esm-ssp585-ocnocean-alk-stop_ (from year 200 onward)**<br>_esm-ssp585ext_ (from year 200 onward)**<br>_esm-ssp585-ocnocean-alk-ext_ (from year 200 onward)** |

*In the _historical_ and _yr2010CO2yr2010co2_ simulations output is needed only to diagnose (at least annually) $CO_2$ emissions.
**This is from scenario year 2300 onward.

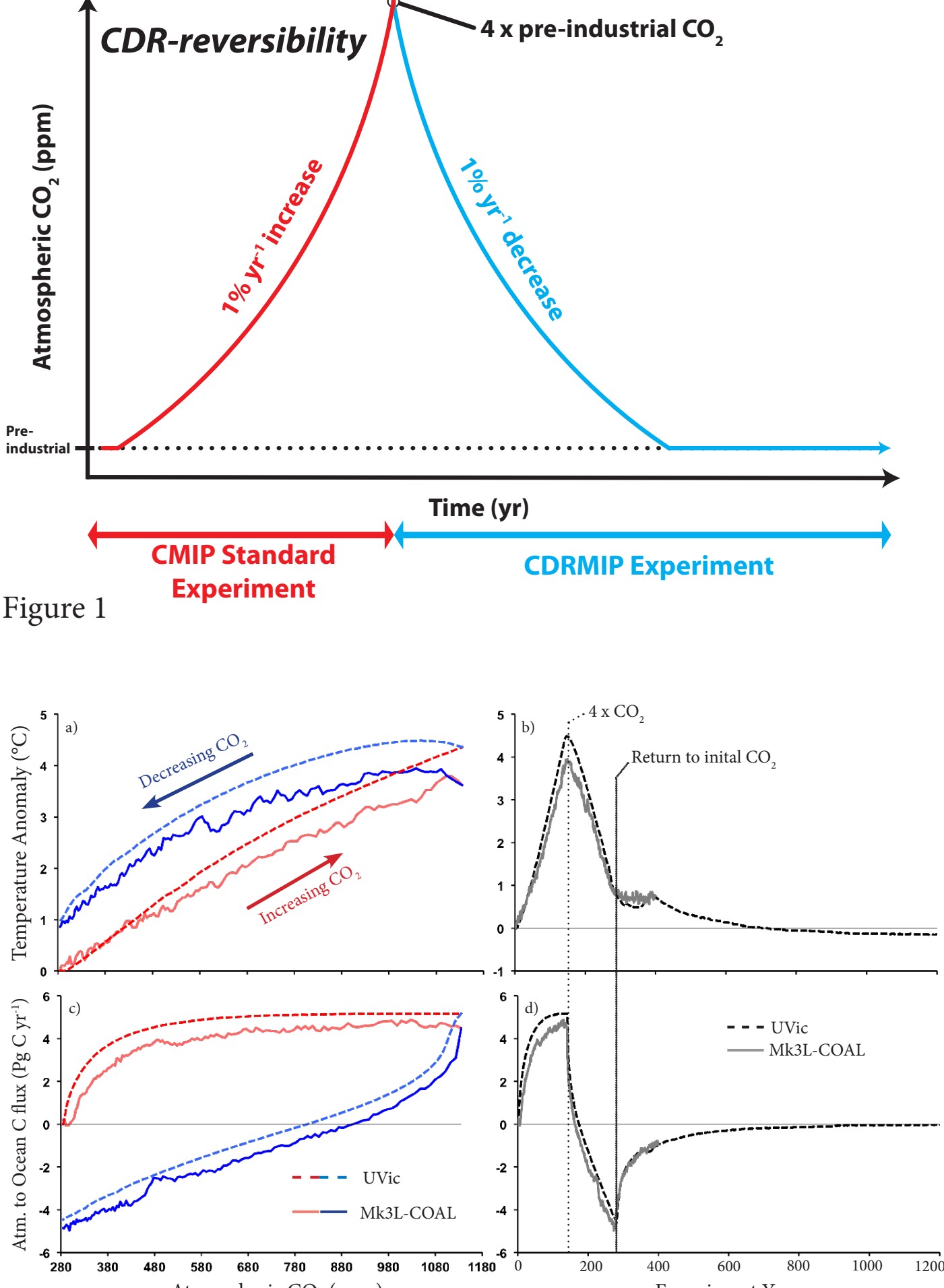

Figure 1

Figure 2

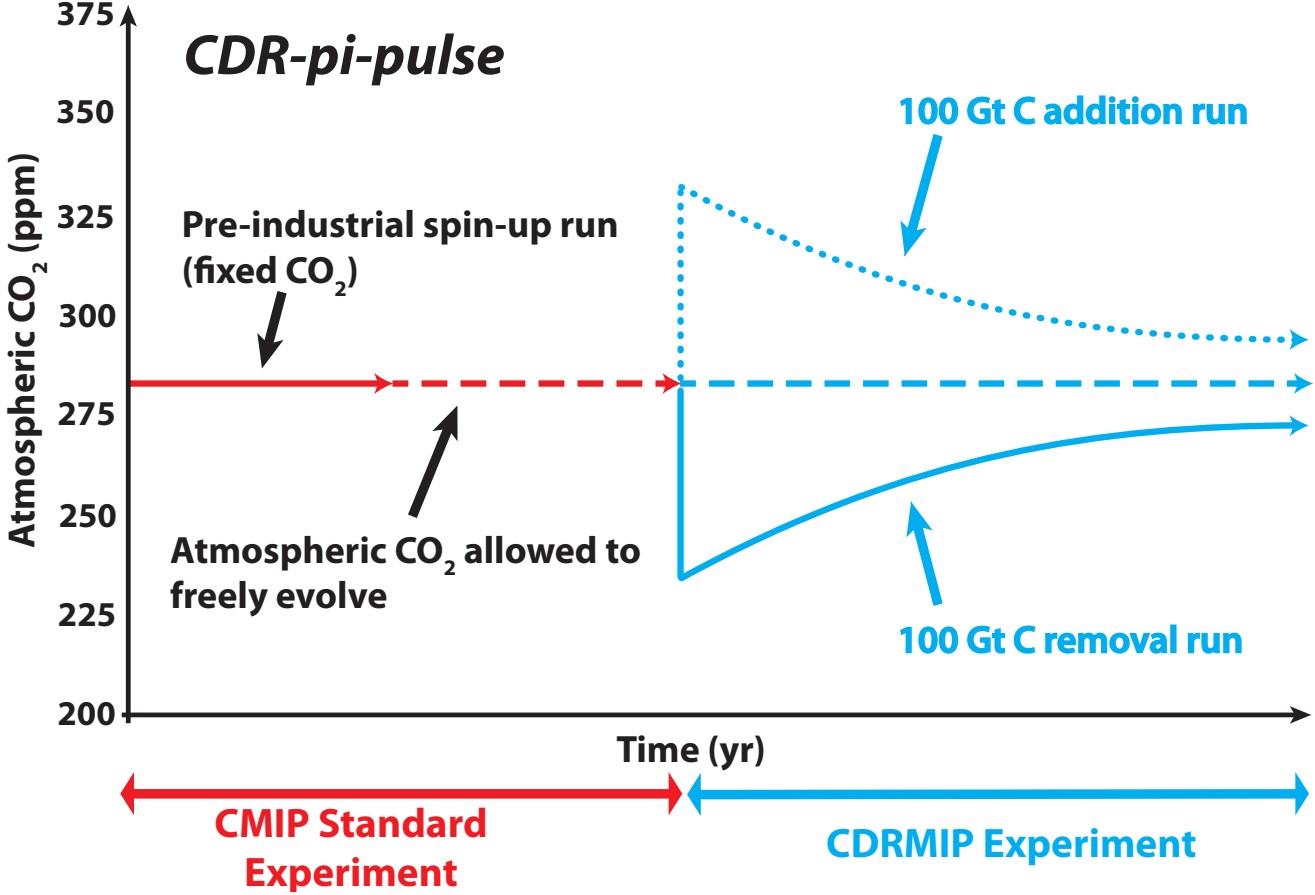

Figure 3

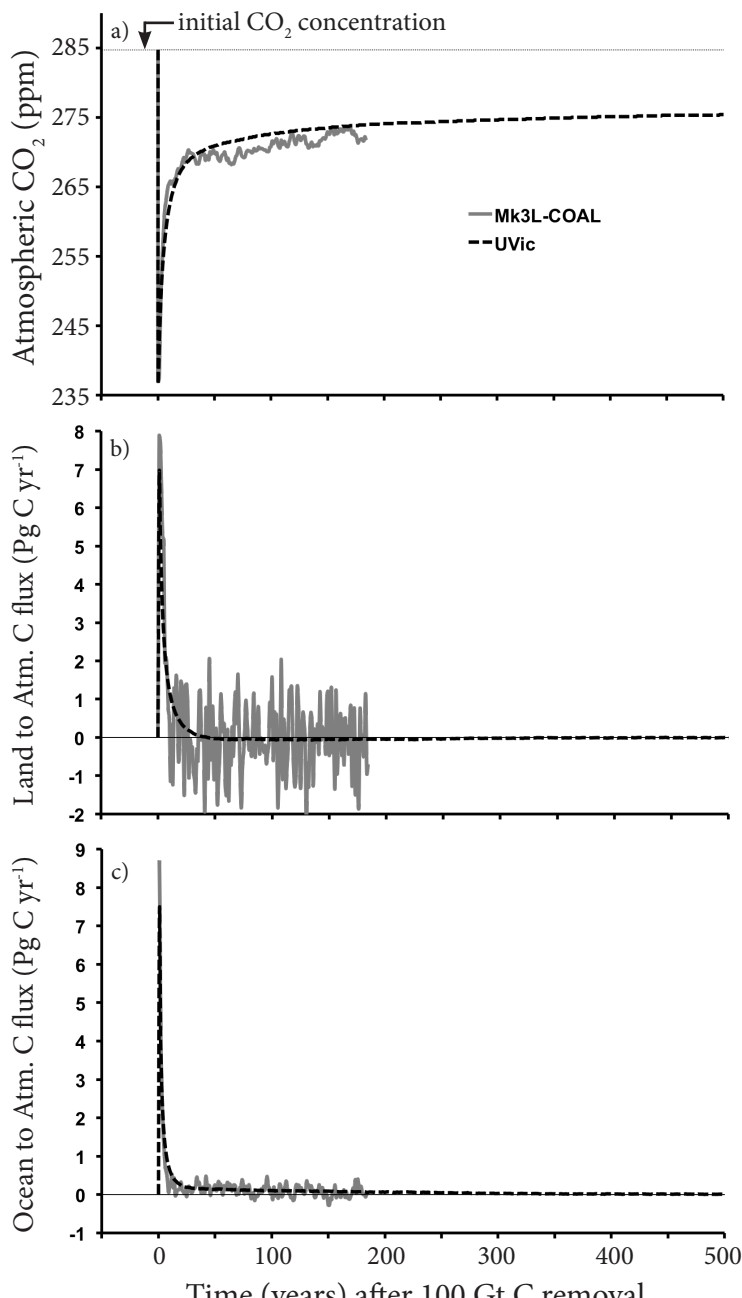

Figure 4

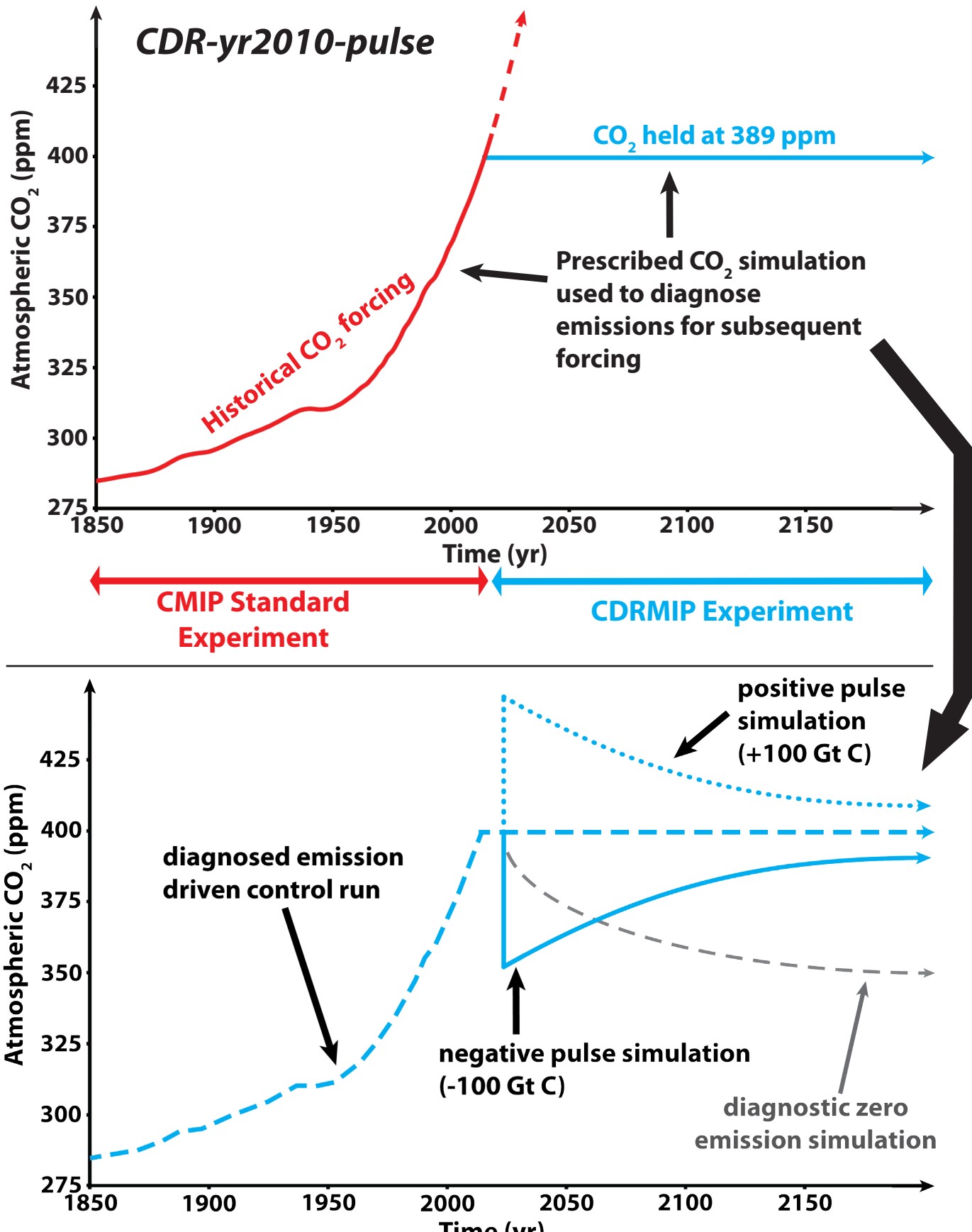

Figure 5

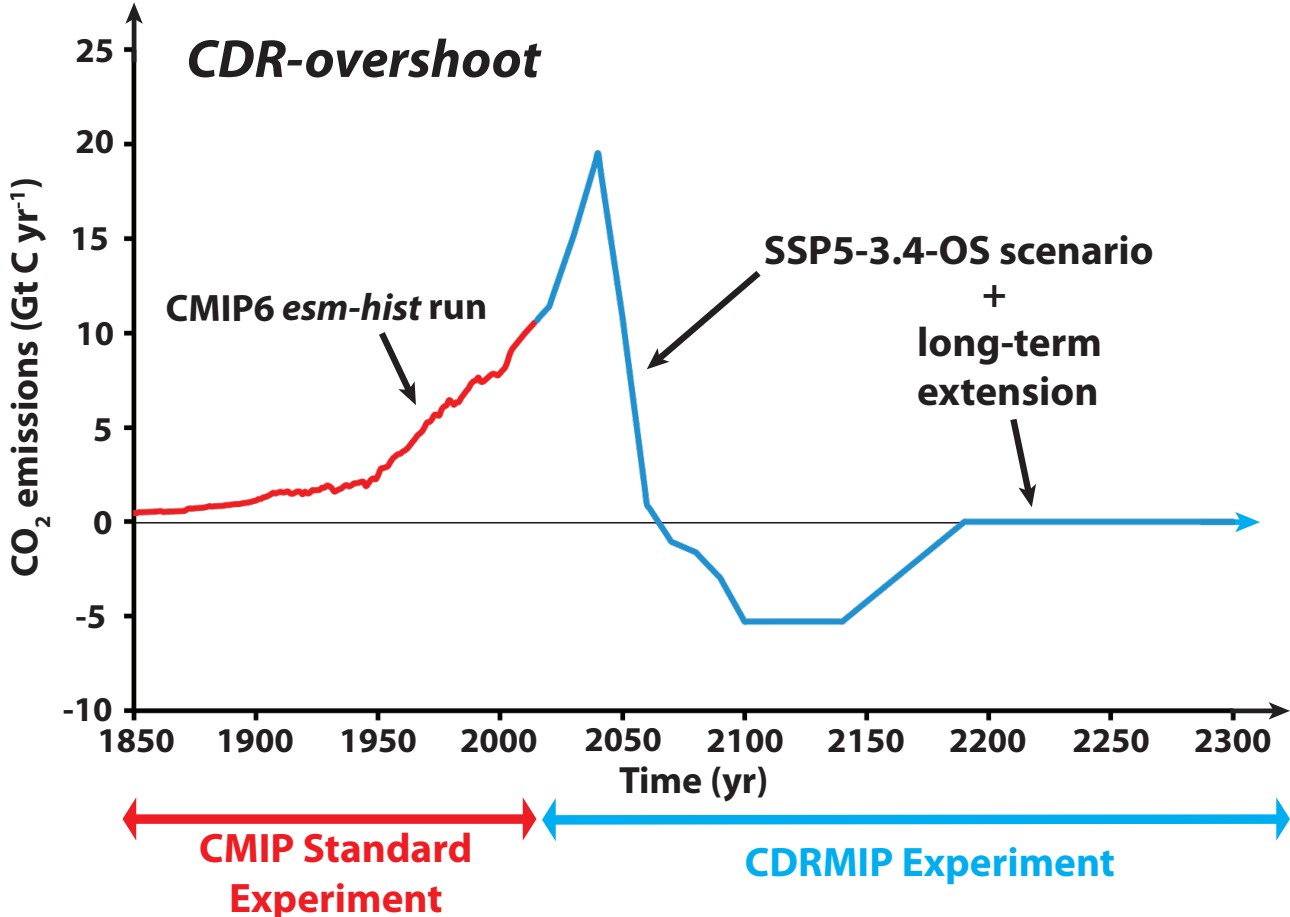

Figure 6

Figure 1. Schematic of the CDRMIP climate and carbon cycle reversibility experimental protocol (*CDR-reversibility*). From a preindustrial run at steady state atmospheric $CO_2$ is prescribed to increase and then decrease over a ~280 year period, after which it is held constant for as long as computationally possible.

Figure 2. Exemplary climate and carbon cycle reversibility experiment (*CDR-reversibility*) results with the Mk3L-COAL Earth system model and the University of Victoria (UVic) Earth system model of intermediate complexity (models are described in Appendix D). The left panels show annual global mean (a) temperature anomalies (°C; relative to pre-industrial temperatures) and (c) the atmosphere to ocean carbon fluxes (Pg C yr$^{-1}$) versus the atmospheric $CO_2$ (ppm) during the first 280 years of the experiment (i.e., when $CO_2$ is increasing and decreasing). The right panels show the same (b) temperature anomalies and (d) the atmosphere to ocean carbon fluxes versus time. Note that the Mk3L-COAL simulation was only 400 years long.

Figure 3. Schematic of the CDRMIP instantaneous $CO_2$ removal / addition from an unperturbed climate experimental protocol (*CDR-pi-pulse*). Models are spun-up for as long as possible with a prescribed preindustrial atmospheric $CO_2$ concentration. Then atmospheric $CO_2$ is allowed to freely evolve for at least 100 years as a control run. The negative / positive pulse experiments are conducted by instantly removing or adding 100 Gt C to the atmosphere of a simulation where the atmosphere is at steady state and $CO_2$ can freely evolve. These runs continue for as long as computationally possible.

Figure 4. Exemplary instantaneous $CO_2$ removal from a preindustrial climate experiment (*CDR-pi-pulse*) results from the *esm-pi-cdr-pulse* simulation with the Mk3L-COAL Earth system model and the University of Victoria (UVic) Earth system model of intermediate complexity (models are described in Appendix D). (a) shows atmospheric $CO_2$ vs. time, (b) the land to atmosphere carbon flux vs. time, and (c) the ocean to atmosphere carbon flux vs. time. Note that the Mk3L-COAL simulation was only 184 years long.

Figure 5. Schematic of the CDRMIP instantaneous $CO_2$ removal / addition from a perturbed climate experimental protocol (*CDR-yr2010-pulse*). Top panel: Initially historical $CO_2$ forcing is prescribed and then held constant at 389 ppm (~ year 2010) while $CO_2$ emissions are diagnosed. Bottom panel: A control simulation is conducted using the diagnosed emissions. The negative / positive pulse experiments are conducted by instantly removing or adding 100 Gt C to the atmosphere of the $CO_2$ emission-driven simulation 5 years after $CO_2$ reaches 389 ppm. Another control simulation is also conduced that sets emissions to zero at the time of the negative pulse. The emission-driven simulations continue for as long as computationally possible.

Figure 6. Schematic of the CDRMIP emission-driven SSP5-3.4-OS scenario experimental protocol (*CDR-overshoot*). A $CO_2$ emission-driven historical simulation is conducted until the year 2015. Then an emission-driven simulation with SSP5-3.4-OS scenario forcing is conducted. This simulation is extended until

the year 2300 using SSP5-3.4-OS scenario long-term extension forcing. Thereafter, runs may continue for as long as computationally possible with constant forcing after the year 2300.