# Peer review of "The Carbon Dioxide Removal Model Intercomparison Project"

_Geoscientific Model Development, 2017_

## Referee Comment (RC1) · Anonymous Referee #1 · 19 Sep 2017

Review Keller et al. 'The Carbon Dioxide Removal Model Intercomparison Project (CDR-MIP): Rationale and experimental design'

This manuscript presents a motivation and description of the experimental design of a planned carbon dioxide removal model intercomparison project. The manuscript touches upon a much discussed but so far little investigated area: how will the Earth system react to large scale removal of carbon from the atmosphere by different processes? This is an important initiative that will serve the community well and I find the article worthy of publication in Geoscientific Model Development. The motivation and experimental protocol is outlined well but for clarity I recommend some changes listed below.

**1 Section 1.2 CDR-MIP Scientific Foci**
[Page 6]The first and second motivation seem to address the same question and could maybe put together.

**2 Section 2 Background and motivation**
[Page 9, lines 270-273] sentence unclear, rephrase
[Page 10, line 315] Maybe shortly name some examples for other side effects than regional albedo changes.

**3 Section 3.1 Relations to other MIPs**
I acknowledge the fact that with the variety of existing MIPs it is not easy to set a new MIP into relation to them. This subsection, however, is generally not very clear to the reader and a bit lengthy with repetitions of statements and needs focusing.

**4 Section 3.5 Model drift**
Shortly state acceptable model drift as described by Jones et al. (2016b) (as done on Page 26, lines 832-839).

**5 Model output frequency subsections in section 4 (4.1.2, 4.2.2, 4.2.4,4.2.6, 4.3.2, 4.4.2)**
Combine these subsections into one and refer to Table 8 for details to avoid extensive repetition.

**6 Section 4.2**
Very lengthy to read. Shorten and focus.

**7 Section 4.2.1**
[Page 26, lines 832-839] move to section 3.5 and remove here.

**8 Section 4.3**
Same as #6, try to shorten and focus.

**9 Section 7 Code and/or data availability**
[Page 41] To avoid repetition, combine this section with section 5.4 into one.

Minor comments

[Page 7, lines 206-207 and 222-225] repetition

[Page 7, lines 223-224] clarify: a good test for what?

[Page 18, line 577] 'not mandatory, nor a prerequisite' replace 'not' with 'neither'.

[Page 19, lines 621-622] In 'limiting the number experiments' add 'of'.

[Pages 20-21, lines 658-661] Remove sentence 'Moreover, since many...'

[Page 21, lines 668-669] Remove sentence 'Note that piControl...'

[Page 28, lines 911-912] Remove sentence 'EMICs and box models...' and include this information in subsection about model output frequency (see #5).

[Page 29, lines 922-924 and 936-937] Remove sentence 'EMICs and box models...'

[Page 45, line 1437] '2.8° longitude by 1.6° longitude' do you mean '2.8° longitude by 1.6° latitude'?

Tables

[Tables 2-7] Including a column with the name of the preceding run from which the experiment is to be started will increase clarity.

---

## Referee Comment (RC2) · B. Sanderson (Referee) · 22 Sep 2017

The submitted paper documents the experimental design for the CDRMIP suite of experiments, designed to explore model uncertainties in Earth System response to climate engineering through potential anthropogenic removal of carbon dioxide from the atmosphere. The MIP is well motivated, and the introduction does a good job of framing why such a MIP would be useful.

The paper should certainly be published, and I look forward to seeing the results of the MIP. I have some minor comments only, which I attach for the authors' consideration.

Minor Comments:

1. The details of the experimental design need clarifying in places. For example, a number of the experiments require 'constant forcing' for non-CO2 agents, but the authors do not explicitly state how to implement this. Should aerosol concentrations be held constant, or should emissions be held constant?

2. There is almost no consideration of internal climate variability, recommended ensemble size, and what role that might have. How many ensemble members are required for each of the experiments to assess the desired signal? If it is only 1, can the authors demonstrate that a single simulation can produce a sufficiently significant result to differentiate the structural differences between different models in the presence of climate noise?

3. It isn't clear how a proposed experiment esm-ssp534-over differs from the existing C4MIP ssp534-over-bgc. Could the authors make this more clear?

4. Could the authors expand on what processes would result in yr2010co2 differing from esm-hist-yr2010co2-control, given that if compatible emissions are correctly diagnosed, they should be identical? The only case, to my mind, where this would not be true is if internally-generated climate noise was capable of changing the compatible emissions requirements. However, if this is the case, then the experimental design is insufficient - and an ensemble of yr2010co2 simulations would be required in order to assess the central estimate for compatible emissions.

5. In esm-hist-yr2010*, what RCP/SSP should be used if 389ppm is not reached during the historical period?

Typos/presentational points:

Line 50: comma after climate Line 118: Do any of the 2 degree scenarios (which have not already diverged from historical emissions) require no CO2 removal? I'm not aware of them. Could they be cited? Line 126: suggest "are not yet a commercial product"

Line 395: This paragraph seems to imply that a GCM can inform policy which differs only in terms of the relative sizes of positive and negative fluxes which make up a net anthropogenic flux. This seems to be true only for a subset of CDR approaches where there are long term consequences of removal for future fluxes (e.g. reforestation), but not really for direct air capture. Perhaps this could be clarified Line 464: Suggest using a word other than "control" here, which is almost universally interpreted as a constant forcing simulations in other CMIP6 MIPs. Line 971: Is esm-535-over-ext a typo?

---

## Referee Comment (RC3) · Anonymous Referee #3 · 30 Sep 2017

In the manuscript 'The Carbon Dioxide Removal Model Intercomparison Project (CDR-MIP): Rationale and experimental design' the authors document the experimental design for a suite of coordinated experiments, designed to explore potential, risks and uncertainties in Earth System response to carbon dioxide removal (CDR) from the atmosphere. The authors provide a sound and detailed motivation for this suite of coordinated experiments, emphasizing connection with other model intercomparison exercises.

I much appreciate this paper which is not only highly relevant in the context of UNFCCC COP21 objectives. IT is also relevant for some WCRP grand challenges topics such

as reducing uncertainties in climate sensitivity and constraining climate-carbon cycle feedbacks. Therefore, I recommend acceptance of this manuscript after some minor revisions listed below.

General comments: 1) Some sectione are really long to read. I would therefore recommend to bring upfront important message. 2) Some experiments seem to complement existing MIP coordinated simulation while some other don't. It would be convenient to clearly state why those later are independent (or new) from existing experiments. 3) There is no documentation or information on how this MIP will address the role of the internal climate variability. As I read the present ms, it seems that exp produce a sufficient signal-to-noise ratio. However, for some exp, especially those in emission-driven simulations recommendation and sensitivity relative to the ensemble size seems required. Specific comments:

L52 Âă: It could be nice somewhere to refer to the IPCC definition of mitigation.

L59 Âă: impacts= climate impacts Âă?; efficacy refer to technological scalability here Âă? I don't think CDR-MIP address this very specific point.

L81 Âă: please indicated what is the reference period used to defined the Âń Âăpreindustrial level Âă Âż

L85 Âă: rather use Âń Âăattributed to anthropogenic. . . Âă Âż L91 Âă: limiting warming= limiting anthropogenic warming

L116 Âă: please indicate that these are all models(=IAMs) results and are hence speculative. . .

L135-141 Âă: Âń Âăhelp to mitigation Âă Âż and Âń Âăpotential effectiveness Âă Âż are redundant. The last point need to be clearer. As I undertstand the various foci of CDR-MIP, there are Âă: - Effectiveness - Risks and benefist including avoided impacts - Related carbon cycle –climate feedbacks

L235 Âă: issue of permanence has to be taken with cautious here. Indeed, CDR-MIP

is designed for ESM, EMIC and boxmodel. Those models are not designed to address carbon storage leakage (fit for purpose). They can only document the response of the Earth system when a leakage occurs.

L273Âă: Please refer to {Smith:2015hg}

L386Âă: CMIP5, are you sureÂă?

L654 doubtful = unrealistic

L663Âă: Why C1 doesn't rely on abrupt 4xCO2 rather than 1%CO2.

L841Âă: As I read itÂă: there is a removal of 100Gt in one year. Are you expecting a pulse removal (1 model time-step) are a smoothed removal during one yearÂă? Besides, do you expect a spatial structure of the CO2 removalÂă?

L1043-1047Âă: Why not relying on a constant afforestationÂă? LUMIP T1 exp is a constant deforestation. It would have been a complementary model experiments here.

L1437 2.8° longitude by 1.6° latitude

---

## Author Comment (AC4) · 4 Dec 2017

In the revised manuscript we have changed how we prioritize the experiments, i.e., the tiers have been changed in some cases. In the original GMDD manuscript we had five tier 1 experiments, one tier 2, and several tier 3 experiments that extended a tier 1 experiment. After careful consideration and discussions with modelling groups participating in CMIP6 we have decided to reduce the number of Tier 1 experiments to two, the C1 and C2_pi-pulse experiments. The main reason for this change is that groups participating in CMIP6 have told us that they are participating in many other MIPs and would be unable to commit to performing five tier 1 experiments, but could

potentially commit to running two CDR-MIP experiments. This does not mean that the former tier 1 experiments, i.e., C2_overshoot, C3, and C4, are not high priorities for CDR-MIP, just that groups do not have to commit to performing five tier 1 experiments. Ideally groups will be able to perform all CDR-MIP experiments.

---

## Author Response (AR1)

Response to reviewer #1

**>> The reviewer's comments are in bold**. **<<**
*>> Responses are in italics. <<*
>> New text is in plain type. <<

**Review Keller et al. 'The Carbon Dioxide Removal Model Intercomparison Project (CDR-MIP): Rationale and experimental design'**

**This manuscript presents a motivation and description of the experimental design of a planned carbon dioxide removal model intercomparison project. The manuscript touches upon a much discussed but so far little investigated area: how will the Earth system react to large scale removal of carbon from the atmosphere by different processes? This is an important initiative that will serve the community well and I find the article worthy of publication in Geoscientific Model Development. The motivation and experimental protocol is outlined well but for clarity I recommend some changes listed below.**

**#1 Section 1.2 CDR-MIP Scientific Foci[Page 6] The first and second motivation seem to address the same question and could maybe put together.**

*Thank you for the suggestion. We agree that they are similar and have combined these motivations (Page 6, lines 186-210).*

**#2 Section 2 Background and motivation[Page 9, lines 270-273] sentence unclear, rephrase**

*Sorry if this is unclear. We have tried to clarify the sentence (Page 9, lines 290-295 by rephrasing it to be:*

" BECCS is thus, constrained by some environmental limitations (e.g., suitable land area), but because the carbon is removed and ultimately stored elsewhere, it may have a higher CDR potential than if the same deployment area were used for a sink-enhancing CDR method like afforestation that stores carbon permanently above ground and reaches a saturation level for a given area."

**[Page 10, line 315] Maybe shortly name some examples for other side effects than regional albedo changes.**

*We have added a few more examples and slightly changed the sentence order so that the order is logical. This section (Page 10, lines 330-343) now reads:*

" Some significant side effects are caused by the spatial scale (e.g., millions of $km^2$) at which many methods would have to be deployed to have a significant impact upon $CO_2$ and global temperatures (Boysen et al., 2016; Heck et al., 2016; Keller et al., 2014). Side effects can also potentially alter the natural environment by disrupting biogeochemical and hydrological cycles, ecosystems, and biodiversity (Keller et al., 2014). For example, large-scale afforestation could change regional albedo and evapotranspiration and so have a biogeophysical impact on the Earth's energy budget and climate (Betts, 2000; Keller et al., 2014). Additionally, if afforestation were done with non-native plants or monocultures to increase carbon removal rates this could impact local biodiversity."

**#3 Section 3.1 Relations to other MIPsI acknowledge the fact that with the variety of existing MIPs it is not easy to set a new MIP into relation to them. This subsection, however, is generally not very clear to the reader and a bit lengthy with repetitions of statements and needs focusing.**

*We have tried to improve this section (Page 14-16, lines 470-608). Hopefully it is now more clear and concise without repetitive statements. The section now reads:*

[revised manuscript text omitted]

**#4 Section 3.5 Model drift - Shortly state acceptable model drift as described by Jones et al. (2016b) (as done on Page 26, lines 832-839).**

*Done. Text has been added (Page 19, lines 702-705) stating that,* "This means that land, ocean and atmosphere carbon stores should each vary by less than 10 GtC per century (long-term average ≤ 0.1 Gt C yr$^{-1}$). We leave it to individual groups to determine the length of the run required to reach such a state."

**#5 Model output frequency subsections in section 4 (4.1.2, 4.2.2, 4.2.4,4.2.6, 4.3.2, 4.4.2) Combine these subsections into one and refer to Table 8 for details to avoid extensive repetition.**

*Thanks for the suggestion. These sections have been combined and placed into a new Section - 5.4 (Pages 34-36, lines 1670-1760).*

**#6 Section 4.2 Very lengthy to read. Shorten and focus.**

*We have deleted two large sections of text that were repetitions of what had been stated in Sections 2 and 3.1. This should shorten and focus the section.*

**#7 Section 4.2.1[Page 26, lines 832-839] move to section 3.5 and remove here.**

*Done.*

**#8 Section 4.3 Same as #6, try to shorten and focus.**

*We have deleted a large section of text to shorten this section down to two, more focused paragraphs.*

**#9 Section 7 Code and/or data availability[Page 41] To avoid repetition, combine this section with section 5.4 into one.**

*We had originally done this, but the journal explicitly requires that we have section on "Code and/or Data Availability", which is why we added this section at the request of the Journal after uploading our original manuscript. However, we do agree that some information is repetitive and have tried to change text in other sections to refer to this one if possible.*

**Minor comments**

**[Page 7, lines 206-207 and 222-225] repetition**

*The sentence that was on lines 206-207 had been deleted to avoid repetition.*

**[Page 7, lines 223-224] clarify: a good test for what?**

*We have deleted this sentence since it repeats, in a less clear manner, what was said in the introductory paragraph to this section (Page 6, lines 202-207) where we state that,* "CDR-MIP results may also be able to provide information that helps to understand how model resolution and complexity cause systematic model bias. In this instance, CDR-MIP experiments may be especially useful for gaining a better understanding of the similarities and differences between global carbon cycle models because we invite a diverse group of models to participate in CDR-MIP".

**[Page 18, line 577] 'not mandatory, nor a prerequisite' replace 'not' with 'neither'.**

*Corrected (Page 18, line 672).*

**[Page 19, lines 621-622] In 'limiting the number experiments' add 'of'.**

*Corrected (Page 19, line 722).*

**[Pages 20-21, lines 658-661] Remove sentence 'Moreover, since many...'**

*Done.*

**[Page 21, lines 668-669] Remove sentence 'Note that piControl...'**

*Done.*

**[Page 28, lines 911-912] Remove sentence 'EMICs and box models...' and include this information in subsection about model output frequency (see #5).**

*Done.*

**[Page 29, lines 922-924 and 936-937] Remove sentence 'EMICs and box models...'**

*Done.*

**[Page 45, line 1437] '2.8° longitude by 1.6° longitude' do you mean '2.8° longitude by 1.6° latitude'?**

*Yes, this has been corrected (Page 43, line 1973).*

**Tables**

**[Tables 2-7] Including a column with the name of the preceding run from which the experiment is to be started will increase clarity.**

*Thanks for the suggestion. A new column called "Initialized using a restart from" has been added to each of these tables.*

Response to reviewer #2

**>> The reviewer's comments are in bold**. **<<**
*>> Responses are in italics. <<*
>> New text is in plain type. <<

Review:

**The submitted paper documents the experimental design for the CDRMIP suite of experiments, designed to explore model uncertainties in Earth System response to climate engineering through potential anthropogenic removal of carbon dioxide from the atmosphere. The MIP is well motivated, and the introduction does a good job of framing why such a MIP would be useful.**
**The paper should certainly be published, and I look forward to seeing the results of the MIP. I have some minor comments only, which I attach for the authors' consideration.**

**Minor Comments:**

**1. The details of the experimental design need clarifying in places. For example, a number of the experiments require 'constant forcing' for non-CO2 agents, but the authors do not explicitly state how to implement this. Should aerosol concentrations be held constant, or should emissions be held constant?**

*Sorry for leaving out these details. We have added a paragraph to Section 4 (Page 20, lines 734-743) to clarify what we mean by constant forcing. This paragraph reads,*

"In some of the experiments described below we ask that non-$CO_2$ forcing (e.g., land use change, radiative forcing from other greenhouse gases, etc.) be held constant, e.g. at that of a specific year, so that only changes in other forcing, like $CO_2$ emissions, drive the main model response. For some forcing, e.g. aerosol emissions, this may mean that monthly changes in forcing are repeated throughout the rest of the simulation as if it was always one particular year. However, we recognize that models apply forcing in different ways and leave it to individual modelling groups to determine the best way hold forcing constant. We request that the methodology for holding forcing constant be documented for each model."

**2. There is almost no consideration of internal climate variability, recommended ensemble size, and what role that might have. How many ensemble members are required for each of the experiments to assess the desired signal? If it is only 1, can the authors demonstrate that a single simulation can produce a sufficiently significant result to differentiate the structural differences between different models in the presence of climate noise?**

*We do recommend that groups conduct 3 ensemble members (Section 3.3 on page 18) to deal with variability. However, for CDR-MIP, interannual variability is likely to be a larger issue than internal model variability. Pervious studies such as Hewitt et al., (2016) that looked at this issue with a focus on the carbon cycle, which is especially relevant for CDR-MIP, found that when comparing simulations of CMIP5 scenarios for land-carbon fluxes, the model spread was so big that it was the primary source of uncertainty. While for ocean carbon uptake, the variance attributed to differences between representative concentration pathway scenarios exceeded the variance attributed to differences between climate models. In most models "internal variability" (assuming this means "sensitivity to perturbed initial conditions") was fairly small – especially on decadal scales. Interannual variability of carbon fluxes was high, but tended to even out on >5 year timescales. Based on this knowledge, we recommend that modelling groups perform at least three ensemble members to reduce this uncertainty related to variability, but leave it up to each group to determine how much of an issue this is and whether it requires more or fewer runs. Thus, section 3.3 states that, "* We encourage participants whose models have internal variability to conduct multiple realizations, i.e. ensembles, for all experiments. While these are highly desirable, they are neither mandatory, nor a prerequisite for participation in CDR-MIP. Therefore, the number of ensemble members is at the discretion of each modeling group. However, we strongly encourage groups to submit at least three ensemble members if possible."

**3. It isn't clear how a proposed experiment esm-ssp534-over differs from the existing C4MIP ssp534-over-bgc. Could the authors make this more clear?**

*The reviewer is likely referring to the statement in section 4.2 where we stated that,*

"*We also highly recommend that groups conduct the ScenarioMIP ssp534-over and ssp534-over-ext and C4MIP ssp534-over-bgc and ssp534-over-bgcExt simulations as these runs will be invaluable for qualitative comparisons.*"

*We agree that the relationship between these simulations was not clear from this isolated statement. We have deleted this statement to avoid repetition (as recommended by reviewer #1) and now highlight the relationship between these simulations in Section 3, where more detail is provided. Here (Page 14, lines 530-537) we state that:*

"*The C4MIP experiment ssp534-over-bgc* is a concentration driven "overshoot" scenario simulation that is run in a partially coupled mode. The simulation required to analyze this experiment is a fully coupled $CO_2$ concentration driven simulation of this scenario, *ssp534-over,* from the Scenario Model Intercomparison Project (ScenarioMIP). The novel CDR-MIP experiment, *C2_overshoot*, which is a fully coupled $CO_2$ emission driven version of this scenario, will provide additional information that can be used to extend the analyses to better understand climate-carbon cycle feedbacks."

*We also have similar statements in Section 3.2 (Page 17, lines 649-666) that read,*

"We also highly recommend that groups run these additional C4MIP and ScenarioMIP simulations:

- The ScenarioMIP *ssp534-over* and *ssp534-over-ext* simulations, which prescribe the atmospheric $CO_2$ concentration to follow an emission overshoot pathway that is followed by aggressive mitigation to reduce emissions to zero by about 2070, with substantial negative global emissions thereafter. These results can be qualitatively compared to CDR-MIP experiment *C2_overshoot,* which is the same scenario, but driven by $CO_2$ emissions.

- The C4MIP *ssp534-over-bgc* and *ssp534-over-bgcExt* simulations, which are biogeochemically-coupled versions of the *ssp534-over* and *ssp534-over-ext* simulations, i.e., only the carbon cycle components (land and ocean) see the prescribed increase in the atmospheric $CO_2$ concentration; the model's radiation scheme sees a fixed preindustrial $CO_2$ concentration. These results can be qualitatively compared to CDR-MIP experiment *C2_overshoot,* which is a fully coupled version of this scenario."

**4. Could the authors expand on what processes would result in yr2010co2 differing from esm-hist-yr2010co2-control, given that if compatible emissions are correctly diagnosed, they should be identical? The only case, to my mind, where this would not be true is if internally-generated climate noise was capable of changing the compatible emissions requirements. However, if this is the case, then the experimental design is insufficient - and an ensemble of yr2010co2 simulations would be required in order to assess the central estimate for compatible emissions.**

*In the test simulations that we have performed with both an ESM and EMIC it appears that climate-carbon cycle feedbacks, which become evident when atmospheric $CO_2$ is allowed to freely evolve, can result in the diagnosed $CO_2$ emissions forcing either slightly under- or overestimating the emissions needed to reach 389ppm.  We agree that in such cases our original design was insufficient and have added text to clarify the necessary steps to achieve the correct atmospheric $CO_2$ concentration.  This text (Page 26, lines 1141-1145) reads,*

"If there are significant differences, e.g., due to climate-carbon cycle feedbacks that become evident when atmospheric $CO_2$ is allowed to freely evolve, then they must be diagnosed and used to adjust the $CO_2$ emission forcing.  In some cases it may be necessary to perform an ensemble of simulations to diagnose compatible emissions."

**5. In esm-hist-yr2010*, what RCP/SSP should be used if 389ppm is not reached during the historical period?**

*For groups performing the CMIP6 historical simulation achieving 389ppm should not be a problem as this is part of the prescribed historical forcing. However, we agree that it could be an issue for those using a CMIP5 model configuration and forcing. We have therefore recommended that they use the RCP 8.5 simulation to reach 389 ppm and the sentence (Page 25, lines 1091-1094) now reads,* " An existing run or setup from CMIP5 or CMIP6 may also be used to reach a $CO_2$ concentration of 389ppm, e.g., the RCP 8.5 CMIP5 simulation or the CMIP6 *historical* experiment."

**Typos/presentational points:**

**Line 50: comma after climate**

*Corrected.*

**Line 118: Do any of the 2 degree scenarios (which have not already diverged from historical emissions) require no CO2 removal? I'm not aware of them. Could they be cited?**

*We are not aware of any of limited warming scenarios without CDR either and have changed the text accordingly. In our original statement we had been referring to scenarios that have already diverged from historical emissions, but now realize that it doesn't make sense to refer to them. The text (Page 4, lines 122-125) is now:* "All Integrated Assessment Model (IAM) scenarios of the future state that some form of CDR will be needed to prevent the mean global surface temperature from exceeding 2°C (Bauer et al., 2017; Fuss et al., 2014; Kriegler et al., 2016; Rogelj et al., 2015a)."

**Line 126: suggest "are not yet a commercial product"**

*Change made (Page 4, line 131).*

**Line 395: This paragraph seems to imply that a GCM can inform policy which differs only in terms of the relative sizes of positive and negative fluxes which make up a net anthropogenic flux. This seems to be true only for a subset of CDR approaches where there are long term consequences of removal for future fluxes (e.g. reforestation), but not really for direct air capture. Perhaps this could be clarified**

*We have clarified this statement to address the issue raised here. The sentence (Page 13, lines 421-425) now reads,* " This relates to the policy relevant question of whether in a regulatory framework, $CO_2$ removals from the atmosphere should be treated like emissions except for the opposite (negative) sign or if specific methods, which may or may not have long-term consequences (e.g., afforestation/reforestation vs. direct $CO_2$ air capture with geological carbon storage), should be treated differently."

**Line 464: Suggest using a word other than "control" here, which is almost universally interpreted as a constant forcing simulations in other CMIP6 MIPs.**

*Done. "control" has been replaced with "simulation". Page 15, line 532.*

**Line 971: Is esm-535-over-ext a typo?**

*Yes, this is a typo and has been corrected.*

Response to reviewer #3

**>> The reviewer's comments are in bold**. **<<**
*>> Responses are in italics. <<*
>> New text is in plain type. <<

**Review:**

**In the manuscript 'The Carbon Dioxide Removal Model Intercomparison Project (CDR- MIP): Rationale and experimental design' the authors document the experimental de- sign for a suite of coordinated experiments, designed to explore potential, risks and uncertainties in Earth System response to carbon dioxide removal (CDR) from the atmosphere. The authors provide a sound and detailed motivation for this suite of coordinated experiments, emphasizing connection with other model intercomparison exercises.**
**I much appreciate this paper, which is not only highly relevant in the context of UNFCCC COP21 objectives. IT is also relevant for some WCRP grand challenges topics such as reducing uncertainties in climate sensitivity and constraining climate-carbon cycle feedbacks. Therefore, I recommend acceptance of this manuscript after some minor revisions listed below.**

**General comments:**

**1) Some sections are really long to read. I would therefore recommend to bring upfront important message.**

*To address this comment and those by other reviewers we have shortened several sections, e.g., Section 3.1, 4.2, and 4.3, and spent a considerable amount of time reducing repetitions, e.g., by condensing the multiple model output frequency sections into one (the new Section 5.4 on Page 34, lines 1676-1763). Hopefully, these improvements have made the text more readable and brought the important messages to the forefront.*

**2) Some experiments seem to complement existing MIP coordinated simulation while some other don't. It would be convenient to clearly state why those later are independent (or new) from existing experiments.**

*As also suggested by another reviewer we have revised the section describing the relationship to other existing MIPs. In doing this we state (Page 14, lines 472-475) up front that, " There are no existing MIPs with experiments focused on climate "reversibility", direct $CO_2$ air capture (with storage), or ocean alkalinization."* *before describing the links that exist between CDR-MIP and other MIPs. This should clarify how CDR-MIP experiments differ from and are complementary to other existing MIP experiments.*

**3) There is no documentation or information on how this MIP will address the role of the internal climate variability. As I read the present ms, it**

**seems that exp produce a sufficient signal-to-noise ratio. However, for some exp, especially those in emission-driven simulations recommendation and sensitivity relative to the ensemble size seems required.**

*We do recommend that groups conduct 3 ensemble members (Section 3.3 on page 18) to deal with variability. However, for CDR-MIP, interannual variability is likely to be a larger issue than internal model variability. Pervious studies such as Hewitt et al., (2016) that looked at this issue with a focus on the carbon cycle, which is especially relevant for CDR-MIP, found that when comparing simulations of CMIP5 scenarios for land-carbon fluxes, the model spread was so big that it was the primary source of uncertainty. While for ocean carbon uptake, the variance attributed to differences between representative concentration pathway scenarios exceeded the variance attributed to differences between climate models. In most models "internal variability" (assuming this means "sensitivity to perturbed initial conditions") was fairly small – especially on decadal scales. Interannual variability of carbon fluxes was high, but tended to even out on >5 year timescales. Based on this knowledge, we recommend that modelling groups perform at least three ensemble members to reduce this uncertainty related to variability, but leave it up to each group to determine how much of an issue this is and whether it requires more or fewer runs. Thus, section 3.3 states that, "*We encourage participants whose models have internal variability to conduct multiple realizations, i.e. ensembles, for all experiments. While these are highly desirable, they are neither mandatory, nor a prerequisite for participation in CDR-MIP. Therefore, the number of ensemble members is at the discretion of each modeling group. However, we strongly encourage groups to submit at least three ensemble members if possible."

**Specific comments (note that in the pdf of original comments the symbols Âă were present):**

**L52: It could be nice somewhere to refer to the IPCC definition of mitigation.**

*We have added the sentence "*To do this a massive climate change mitigation effort to reduce the sources or enhance the sinks of greenhouse gases (IPCC, 2014b) must be undertaken." *to the second paragraph (Page 3, lines 96-98) in the introduction.*

**L59: impacts= climate impacts?; efficacy refer to technological scalability here? I don't think CDR-MIP address this very specific point.**

*Page 2, lines 59-60, "Impacts" has been changed to "climate impacts". No we did not mean efficacy from a technical viewpoint. To clarify what CDR-MIP focuses on we have added text to point out that we are referring to, "*atmospheric $CO_2$ reduction efficacy".

**L81: please indicated what is the reference period used to defined the preindustrial level.**

*We are referring to the year 1850 and have added this information to the sentence (Page 3, line 84).*

**L85: rather use "attributed to anthropogenic...".**

*We have added the words "attributed to" to this sentence (Page 3, line 88).*

**L91: limiting warming= limiting anthropogenic warming**

*Change made, Page 3, line 94.*

**L116: please indicate that these are all models(=IAMs) results and are hence speculative. . .**

*Done, we now state (Page 4, line 122) that "All future Integrated Assessment Model (IAM) scenarios of the future state that...".*

**L135-141: "help to mitigation" and "potential effectiveness" are redundant. The last point need to be clearer. As I understand the various foci of CDR-MIP, there are: - Effectiveness - Risks and benefits including avoided impacts - Related carbon cycle –climate feedbacks**

*We have eliminated the redundant bit from point (ii; line 151) by deleting the word "effectiveness". We have also tried to clarify point (iii; lines 153-156) by changing it to read, " To inform how climate and carbon cycle responses to CDR could be included when calculating and accounting for the contribution of CDR in mitigation scenarios, i.e., so that CDR is better constrained when it is included in IAM generated scenarios."*

**L235: issue of permanence has to be taken with cautious here. Indeed, CDR-MIP is designed for ESM, EMIC and boxmodel. Those models are not designed to address carbon storage leakage (fit for purpose). They can only document the response of the Earth system when a leakage occurs.**

*Yes, thanks for pointing this out as it is an issue. In some models permanence cannot really be calculated. However, for models with more complex components some questions about permanence can be evaluated. For example, if a forest is planted and takes up carbon (afforestation forcing), and then at some point experiences dieback or carbon loss due to a warmer drier future climate (as internally calculated), some of the sequestered carbon may be released again. Or if we add alkalinity to the ocean and then stop adding it at some point, we can evaluate if any of the carbon that was sequestered is released again. We have added a statement to address this issue. Question 4 (Page 7, lines 254-255) now reads, " For methods that enhance natural carbon uptake, e.g., afforestation or ocean alkalinization, where is the carbon stored (land and ocean) and for how long (i.e. issues of permanence; at least as much as this can be calculated with these models)?"*

**L273: Please refer to {Smith:2015}**

*Done.*

**L386: CMIP5, are you sure?**

*Yes, at least some of them are.*

**L654 doubtful = unrealistic**

*Word substitution made (Page 21, line 765).*

**L663: Why C1 doesn't rely on abrupt 4xCO2 rather than 1%CO2.**

*We considered several designs for C1 such as an abrupt 4xCO2 perturbation. However, after much discussion we decided upon a 1%CO2 experiment because it will better capture the slow ocean response to perturbations.*

**L841: As I read it: there is a removal of 100Gt in one year. Are you expecting a pulse removal (1 model time-step) are a smoothed removal during one year? Besides, do you expect a spatial structure of the CO2 removal?**

*Thank you for pointing out that we missed these details. This is an instantaneous removal of $CO_2$. We do not expect a spatial structure for the $CO_2$ removal and will leave it up to modelling groups where $CO_2$ is spatially distributed to find the best way to uniformly remove $CO_2$ from their atmosphere. We have added text so that this section (Page 24, lines 1012-1014) now reads, "* with 100 Gt C instantaneously (within 1 time step) removed from the atmosphere in year 10. If models have $CO_2$ spatially distributed throughout the atmosphere, we suggest removing this amount in a uniform manner."

**L1043-1047: Why not relying on a constant afforestation? LUMIP T1 exp is a constant deforestation. It would have been a complementary model experiments here.**

*We had considered doing such a simulation in our numerous discussions on how to devise an afforestation simulation for CDR-MIP. The main reason that we did not do an afforestation simulation to compliment the LUMIP deforestation simulation is that the deforestation simulation is $CO_2$ concentration-driven and we wanted to have a $CO_2$ emission-driven simulation so that we could quantify climate-carbon cycle feedbacks. The* esm-ssp585- ssp126Lu *was then our best choice, especially since other groups would be performing emission-driven SSP5-8.5 simulations as part of C4MIP and ScenarioMIP.*

**L1437 2.8◦ longitude by 1.6◦ latitude**

*Typo corrected.*

[revised manuscript text omitted]

SCOTT Vivian 20.11.2017 10:37

SCOTT Vivian 20.11.2017 11:00

SCOTT Vivian 20.11.2017 11:01

SCOTT Vivian 20.11.2017 11:01

SCOTT Vivian 20.11.2017 11:01

David Keller 22.11.2017 17:32

David Keller 22.11.2017 17:32

David Keller 24.11.2017 12:03

David Keller 22.11.2017 17:32

David Keller 22.11.2017 17:32

SCOTT Vivian 20.11.2017 11:05

SCOTT Vivian 20.11.2017 11:08

SCOTT Vivian 20.11.2017 11:08

SCOTT Vivian 20.11.2017 11:09

SCOTT Vivian 20.11.2017 11:09

SCOTT Vivian 20.11.2017 11:10

SCOTT Vivian 20.11.2017 11:10

SCOTT Vivian 20.11.2017 11:10

SCOTT Vivian 20.11.2017 11:11

SCOTT Vivian 20.11.2017 11:12

[revised manuscript text omitted]

David Keller 22.11.2017 10:18

David Keller 22.11.2017 10:20
Moved down [1]: 4.1.2 Model output frequency for experiment *C1* .

David Keller 22.11.2017 14:53

David Keller 22.11.2017 10:18

David Keller 24.11.2017 12:32
Deleted: As mentioned in Section 2, the land and ocean components of the carbon cycle will respond to any changes in atmospheric $CO_2$. These reservoirs, which are currently carbon sinks, will oppose any effort to simply remove atmospheric $CO_2$ by either taking up less carbon or by becoming carbon sources to the atmosphere if enough carbon is removed (Jones et al., 2016a; Tokarska and Zickfeld, 2015; Vichi et al., 2013). The carbon cycle is also strongly affected by the climate (Friedlingstein and Prentice, 2010) and thus, its response to DAC will also depend on the past and present state of the climate. These climate-carbon cycle feedbacks make it difficult to determine exactly how much DAC would be needed to reach a specific atmospheric $CO_2$ or temperature target. Only a few modelling studies have investigated how the climate and carbon cycle respond to DAC (Cao and Caldeira, 2010; Jones et al., 2016a; Tokarska and Zickfeld, 2015) and there is much uncertainty that needs to be overcome before quantitative estimates of DAC efficacy can be made.

been proven useful for diagnosing carbon cycle and climate feedbacks in response to $CO_2$ perturbations. For example, previous positive $CO_2$ pulse experiments have been used to calculate Global Warming Potential (GWP) and

Global Temperature change Potential (GTP) metrics (Joos et al., 2013). The experiments described below build upon the previous positive $CO_2$ pulse experiments, i.e., the PD100 and PI100 impulse experiments described in Joos et.

al. (2013) where 100 Gt C is instantly added to preindustrial and near present day simulated climates. However, our experiments also prescribe a negative CDR

pulse as opposed to just adding $CO_2$ to the atmosphere. Two experiments are desirable because the Earth system response to $CO_2$ removal will be different when starting from an equilibrium state versus starting from a perturbed state (Zickfeld et al., 2016). One particular goal of these experiments is to estimate a

Global Cooling Potential (GCP) metric based on a CDR Impulse Response

Function ($IRF_{CDR}$). Such a metric will be useful for calculating how much $CO_2$ is removed by DAC and how much DAC is needed to achieve a particular climate target.

The third experiment, which focuses on "negative emissions", is based on the Shared Socio-economic Pathway (SSP) 5-3.4-overshoot scenario and its long- term extension (Kriegler et al., 2016; O'Neill et al., 2016). This scenario is of interest to CDR-MIP because after an initially high level of emissions, which follows the SSP5-8.5 unmitigated baseline scenario until 2040, $CO_2$ emissions are rapidly reduced with net $CO_2$ emissions becoming negative after the year 2070

and continuing to be so until the year 2190 when they reach zero.  In the original

SSP5-3.4-OS scenario, the negative emissions are achieved using BECCS.

However, as stated earlier there is currently no practical way to design a good multi-model BECCS experiment. Therefore, in our experiments negative emissions are achieved by simply removing $CO_2$ from the atmosphere and assuming that it is permanently stored in a geological reservoir. While this may violate the economic assumptions underlying the scenario, it still provides an opportunity to explore the response of the climate and carbon cycle to potentially achievable levels of negative emissions.

According to calculations done with a simple climate model, MAGICC

version 6.8.01 BETA (Meinshausen et al., 2011; O'Neill et al., 2016), the SSP5-3.4-

OS scenario considerably overshoots the 3.4 W m$^{-2}$ forcing level, with a peak global mean temperature of about 2.4° C, before returning to 3.4 W m$^{-2}$ at the end of the century. Eventually in the long-term extension of this scenario, the forcing stabilizes just above 2 W m$^{-2}$, with a global mean temperature that should equilibrate at about 1.25° C above pre-industrial temperatures. Thus, in addition to allowing an investigation into the response of the climate and carbon cycle to negative emissions, this scenario also provides the opportunity to investigate issues of reversibility, albeit on a shorter timescale and with less of an

"overshoot" than in experiment *C1*.

**4.2.1 Instantaneous CO$_2$ removal / addition from an unperturbed climate**

**experimental protocol (*C2_pi-pulse*)**

This idealized Tier 1 experiment is designed to investigate how the Earth system responds to DAC when perturbed from an equilibrium state (Fig. 3, Table

3). The idea is to provide a baseline system response that can later be compared to the response of a perturbed system, i.e., experiment *C2_yr2010-pulse* (Section

4.2.3). By also performing another simulation where the same amount of CO$_2$ is added to the system, it will be possible to diagnose if the system responds in an inverse manner when the CO$_2$ pulse is positive. Many modelling groups will have already conducted the prerequisite simulation for this experiment in preparation for other modelling research, e.g., during model spin-up or for CMIP, which should minimize the effort needed to perform the complete experiment. The protocol is as follows:

*Prerequisite simulation* - Control simulation under preindustrial conditions with freely evolving CO$_2$. All boundary conditions (solar forcing, land use, etc.) are expected to remain constant. This is also the CMIP5 *esmControl* simulation (Taylor et al., 2012) and the CMIP6 *esm-piControl* simulation (Eyring et al.,

2016). Note that this is exactly the same as PI100 run 4 in Joos et. al. (2013).

*esm-pi-cdr-pulse* simulation - As in *esm-Control* or *esm-piControl*, but with 100 Gt

C instantaneously (within 1 time step) removed from the atmosphere in year 10.

David Keller 24.11.2017 12:10

David Keller 1.11.2017 16:28
Deleted: For groups that have not participated in CMIP5 or CMIP6, this run essentially represents an equilibrium model state with no significant drift. We realize that it is difficult for ESMs to reach a state with little drift and follow the guidelines provided by Jones et al. (2016b), to define what is an acceptably small level of drift in a properly spun-up model, e.g., 
[revised manuscript text omitted]